# Osmostress enhances activating phosphorylation of Hog1 MAP kinase by mono-phosphorylated Pbs2 MAP2K

Kazuo Tatebayashi[1,2,3,*] , Katsuyoshi Yamamoto[2], Taichiro Tomida[4], Akiko Nishimura[2,3], Tomomi Takayama[2,3], Masaaki Oyama[5], Hiroko Kozuka-Hata[5], Satomi Adachi-Akahane[4], Yuji Tokunaga[6] & Haruo Saito[2,**]

## Abstract

The MAP kinase (MAPK) Hog1 is the central regulator of osmoadaptation in yeast. When cells are exposed to high osmolarity, the functionally redundant Sho1 and Sln1 osmosensors, respectively, activate the Ste11-Pbs2-Hog1 MAPK cascade and the Ssk2/Ssk22-Pbs2-Hog1 MAPK cascade. In a canonical MAPK cascade, a MAPK kinase kinase (MAP3K) activates a MAPK kinase (MAP2K) by phosphorylating two conserved Ser/Thr residues in the activation loop. Here, we report that the MAP3K Ste11 phosphorylates only one activating phosphorylation site (Thr-518) in Pbs2, whereas the MAP3Ks Ssk2/Ssk22 can phosphorylate both Ser-514 and Thr-518 under optimal osmostress conditions. Mono-phosphorylated Pbs2 cannot phosphorylate Hog1 unless the reaction between Pbs2 and Hog1 is enhanced by osmostress. The lack of the osmotic enhancement of the Pbs2-Hog1 reaction suppresses Hog1 activation by basal MAP3K activities and prevents pheromone-to-Hog1 crosstalk in the absence of osmostress. We also report that the rapid-and-transient Hog1 activation kinetics at mildly high osmolarities and the slow and prolonged activation kinetics at severely high osmolarities are both caused by a common feedback mechanism.

**Keywords** HOG pathway; Hog1; MAP kinase; osmostress; signal transduction
**Subject Categories** Post-translational Modifications & Proteolysis; Signal Transduction
**The EMBO Journal (2020) 39: e103444**

## Introduction

The family of mitogen-activated protein kinases (MAPKs) are major intracellular signal transducers in eukaryotic cells and are associated with many human diseases (Chen *et al*, 2001; Dhanasekaran & Johnson, 2007). Each MAPK is activated in a three-tiered kinase cascade composed of a MAPK kinase kinase (MAPKKK or MAP3K), a MAPK kinase (MAPKK or MAP2K), and a MAPK. In the canonical model of the MAPK cascades, an activated MAP3K activates a cognate MAP2K by phosphorylating two conserved serine (Ser) and/or threonine (Thr) residues in the flexible activation loop of the MAP2K. In turn, an activated MAP2K activates a cognate MAPK by phosphorylating the conserved Thr and tyrosine (Tyr) residues in the latter's activation loop.

MAPK cascades are highly conserved from yeast to mammalian species, so much so that the mammalian MAPK p38 can functionally complement the yeast MAPK Hog1 (Han *et al*, 1994). A MAPK signal transduction pathway commonly comprises, in addition to the core MAPK cascade, an upstream transmembrane receptor or sensor that detects specific extracellular stimuli, and downstream MAPK substrate molecules (effectors) both in the cytoplasm and in the nucleus. Several different MAPK pathways often co-exist within a cell. In yeast, for example, four MAPKs (Slt2/Mpk1, Kss1, Fus3, and Hog1) are expressed in a cell (Gustin *et al*, 1998). If inappropriate crosstalk occurred between two MAPK cascades, a stimulus aimed at activation of only one of these cascades could incite irrelevant or even detrimental responses.

Different MAPKs in a species are highly homologous to each other, and so are MAP2Ks. Thus, prevention of inappropriate crosstalk between MAPK cascades requires elaborate mechanism for any MAPK cascade, but its difficulty can be most clearly exemplified by the MAPK cascades in yeast, in which three different MAPK

1 Laboratory of Molecular Genetics, Frontier Research Unit, Institute of Medical Science, The University of Tokyo, Tokyo, Japan
2 Division of Molecular Cell Signaling, Institute of Medical Science, The University of Tokyo, Tokyo, Japan
3 Department of Biological Sciences, Graduate School of Science, The University of Tokyo, Tokyo, Japan
4 Department of Physiology, School of Medicine, Faculty of Medicine, Toho University, Tokyo, Japan
5 Medical Proteomics Laboratory, Institute of Medical Science, The University of Tokyo, Tokyo, Japan
6 Molecular Profiling Research Center for Drug Discovery, National Institute of Advanced Industrial Science and Technology, Tokyo, Japan
*Corresponding author. Tel: +81 3 5449 5479; E-mail: tategone@ims.u-tokyo.ac.jp
**Corresponding author. Tel: +81 3 5449 5479; E-mail: h-saito@ims.u-tokyo.ac.jp

cascades with different specificities use the same MAP3K Ste11. The MAPK Hog1 is activated by hyperosmotic stress through the high-osmolarity glycerol (HOG) pathway and orchestrates an array of osmoadaptive changes in transcription, translation, cell cycle, and metabolism (Brewster *et al*, 1993; Saito & Posas, 2012; Hohmann, 2015). The current widely held model of the HOG pathway is as follows (Fig 1A). The upstream portion of the HOG pathway comprises the functionally redundant SHO1 and SLN1 branches. In the SHO1 branch, osmosensing complexes composed of Sho1, Opy2, Hkr1, and Msb2 activate the MAP3K Ste11 (Tanaka *et al*, 2014; Tatebayashi *et al*, 2015; Nishimura *et al*, 2016; Yamamoto *et al*, 2016). In the SLN1 branch, the Sln1-Ypd1-Ssk1 phospho-relay mechanism activates the functionally redundant MAP3Ks Ssk2 and Ssk22 (Ssk2/22) (Posas *et al*, 1996). Activated Ste11 and Ssk2/22 are believed to phosphorylate the MAP2K Pbs2 at Ser-514 and Thr-518 (S514 and T518). Phosphorylated Pbs2 then activates Hog1 (Maeda *et al*, 1995; Posas & Saito, 1997).

Two other yeast MAPKs Fus3/Kss1 are activated by the mating pheromones through Ste11 and the MAP2K Ste7 (Bardwell, 2005). Although the mating pheromones activate Ste11, they do not activate Hog1 (Posas & Saito, 1997). Commonly, the absence of phero-mone-to-Hog1 crosstalk is explained by the pathway insulation model, which posits that a scaffold protein holds several components of one pathway close together, so that signal flows only within that pathway (Harris *et al*, 2001). To prevent crosstalk, however, the scaffold proteins must hold kinases for significantly longer than the half-lives of their activities, which could be several minutes or longer. Because scaffold complexes are typically not so stable (Zalatan *et al*, 2012), additional mechanisms other than scaffolding of signaling complexes are likely to be necessary to effectively prevent crosstalk. Here, we report a mechanism that prevents the pheromone-to-Hog1 crosstalk and also suppresses non-specific Hog1 activation by the basal activities of the upstream MAP3Ks.

# Results

### Osmostress can activate the Hog1 MAPK in the absence of the upstream osmosensors

Several studies have reported that Hog1 can be activated at very high osmolarity (> 1 M NaCl) in strains that are defective in both the SLN1 and SHO1 branches, such as *ssk1Δ ste11Δ* and *ssk2/22Δ sho1Δ* (Van Wuytswinkel *et al*, 2000; O'Rourke & Herskowitz, 2004; Zhi *et al*, 2013; Vázquez-Ibarra *et al*, 2019). Since each of the strains used in those studies expressed at least one MAP3K in the HOG pathway (Ssk2, Ssk22, or Ste11), the results were interpreted as evidence for an alternative mechanism for MAP3K activation following osmostress. However, no such MAP3K activating mechanism has been identified.

To examine if Hog1 could be activated by osmostress in the absence of known upstream osmosensor signaling, we constructed a mutant yeast strain that lacked all four transmembrane proteins involved in the SHO1 branch (Sho1, Opy2, Hkr1, and Msb2) as well as the two MAP3Ks essential for the SLN1 branch (Ssk2 and Ssk22). The genotype of this strain, *sho1Δ opy2Δ hkr1Δ msb2Δ ssk2Δ ssk22Δ*, will be abbreviated hereafter as ΔS/O/H/M *ssk2/*

*22Δ*. We measured osmostress-induced activation/phosphorylation of Hog1 using the anti-phospho-p38 immunoblotting assay (Tatebayashi *et al*, 2003) or a Phos-tag band-shift assay (English *et al*, 2015). Immunoblotting assays indicated that exposure of the ΔS/O/H/M *ssk2/22Δ* mutant strain to stronger osmostress (1 M NaCl) indeed induced weak Hog1 phosphorylation (Fig 1B, lanes 11–12). The fact that Hog1 was not phosphorylated in a *pbs2Δ ssk2/22Δ* strain (Fig 1B, lanes 5–8) indicated that the MAP2K Pbs2 was necessary for Hog1 phosphorylation at 1 M NaCl in the absence of upstream osmosensor signaling. A more detailed analysis of the Hog1 NaCl dose–response using the Phos-tag band-shift assay revealed that Hog1 was weakly phosphorylated between 0.8 M and 1.6 M NaCl in the ΔS/O/H/M *ssk2/22Δ* mutant strain (Fig 1C).

Hog1 phosphorylation in ΔS/O/H/M *ssk2/22Δ* was completely abolished by deletion of *STE11* (Fig 1D; compare lanes 2 and 4 in the longer exposure). Conversely, it was greatly enhanced by the presence of a constitutively active Ste11 such as Ste11-Q301P, Ste11-S281D/S285D/T286D (DDD), or Ste11-T596I (van Drogen *et al*, 2000; Tatebayashi *et al*, 2006) (Fig 1D, lanes 5–10). It has been previously observed that the endogenous-level expression of constitutively active Ste11 mutant does not activate Hog1 unless osmostress is applied (Lamson *et al*, 2006; Tatebayashi *et al*, 2006). Using the ΔS/O/H/M *ssk2/22Δ STE11-Q301P* strain, we determined the dose–response of Hog1 phosphorylation at 5 min at various NaCl concentrations (Fig 1E). The observed dose–response clearly differed from the dose–responses of the wild-type (WT) strain (Fig 1F), a SHO1 branch-only strain (*ssk2/22Δ*; Fig 1G), or an SLN1 branch-only strain (*ste11Δ*; Fig 1H). The extents of Hog1 phosphorylation in these strains at 5 min are summarized in Fig 1I.

### Enhancement of the Pbs2-Hog1 reaction by osmostress involves a genuine osmosensing mechanism

That the ΔS/O/H/M *ssk2/22Δ STE11-Q301P* mutant cell that lacks both the SHO1 and SLN1 branches could activate Hog1 in response to osmostress suggested that there may be a previously undefined sensing mechanism that is distinct from both the Sho1 and Sln1 osmosensors. To determine if phosphorylation of Hog1 in ΔS/O/H/M *ssk2/22Δ STE11-Q301P* was a specific reaction to NaCl or a general reaction to osmostress, we examined if Hog1 could be activated not only by NaCl but also by the non-ionic sorbitol. At very high concentrations, two solutions with the same osmolar concentrations (e.g., 1 M NaCl and 2 M sorbitol) do not necessarily have the same osmotic pressure. However, when compared at the same osmotic pressures expressed in M pascal (MPa) units (Fig EV1A), NaCl and sorbitol induced Hog1 phosphorylation almost identically in WT cells (Fig EV1B). Hog1 phosphorylation in the ΔS/O/H/M *ssk2/22Δ STE11-Q301P* mutant cells was also similar in response to NaCl and sorbitol (Fig EV1C and D), indicating that the Hog1 phosphorylation in the absence of the upstream osmosensors involved a genuine osmosensing mechanism. These results indicated that in the HOG signaling pathway, osmostress acts not only at the level of the upstream osmosensors (Sho1 and Sln1), but also at a point downstream of MAP3Ks. Therefore, we conclude that there is a downstream osmosensor distinct from the upstream osmosensors.

**A**

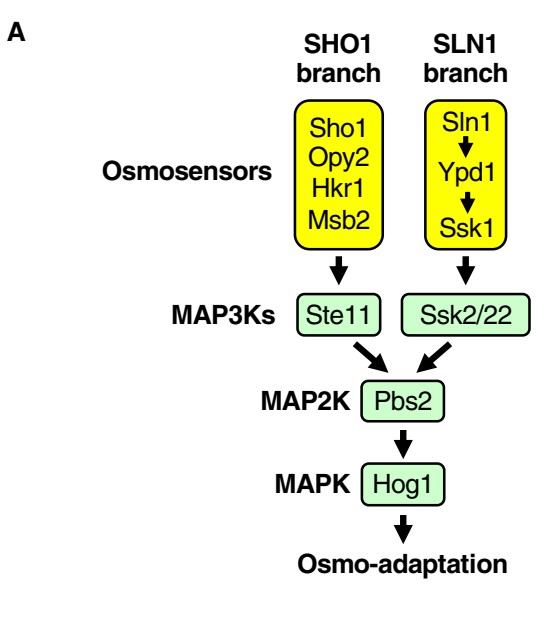

**B**

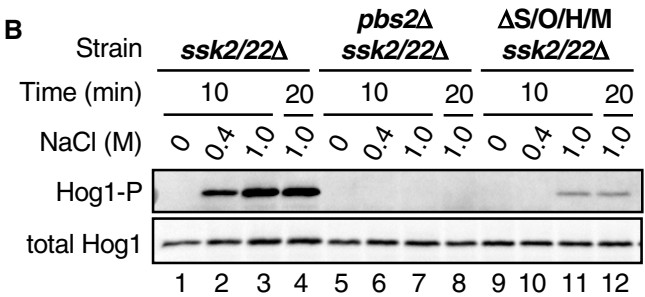

**C**

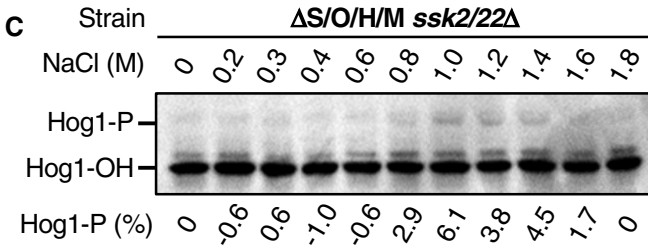

**D**

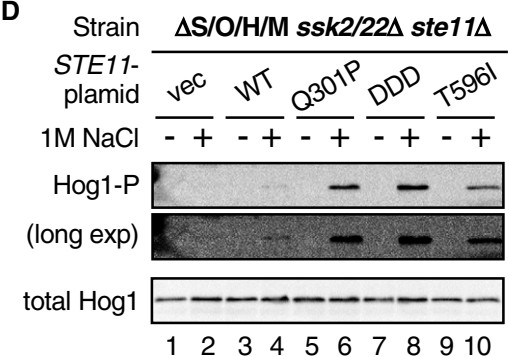

**E**

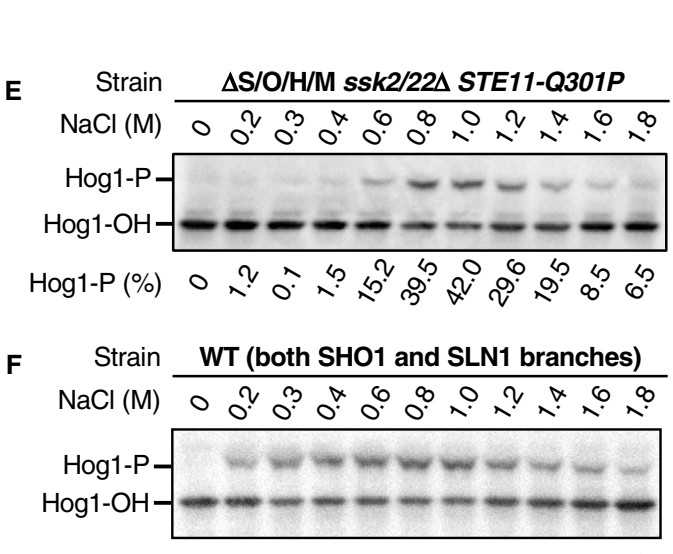

**F**

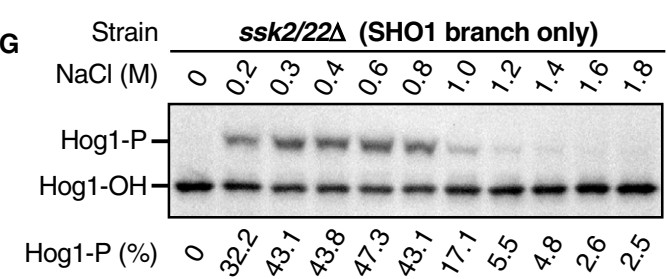

**G**

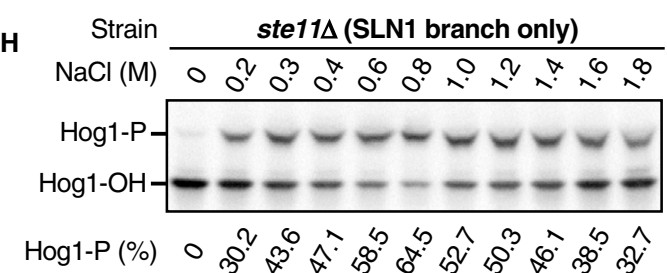

**H**
**I**

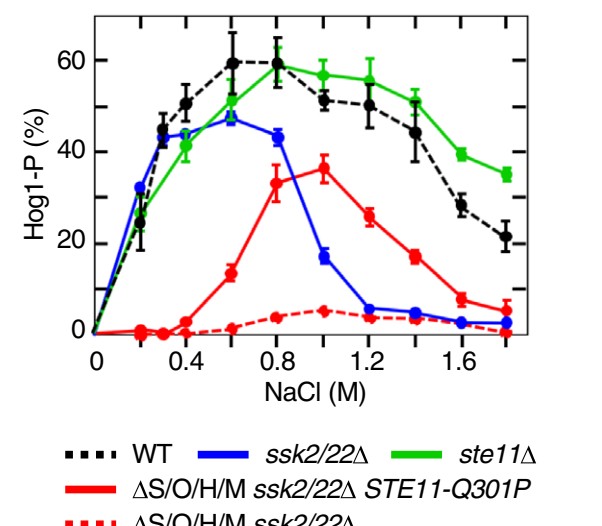

Figure 1.

◀

**Figure 1. Phosphorylation of Hog1 by osmostress in the absence of the upstream osmosensors.**

A   A schematic diagram of the Hog1 MAPK signaling pathway.

B   Analyses of Hog1 phosphorylation by immunoblotting with anti-phospho-p38 (Hog1-P) and anti-Hog1 (total Hog1) antibodies. Cells of the indicated genotypes were stimulated with the indicated concentrations of NaCl for the indicated time. Strains used are TM257, KT207, and KY594-1.

C   Analyses of Hog1 phosphorylation by Phos-tag band-shift assay. Yeast strain KY594-1 was stimulated with the indicated concentrations of NaCl for 5 min. The percentages of phosphorylated Hog1 (Hog1-P [%]) were calculated as explained in Materials and Methods and are shown beneath the panel.

D   Analyses of Hog1 phosphorylation by immunoblotting with anti-phospho-p38 (Hog1-P) and anti-Hog1 (total Hog1) antibodies. Yeast strain KT219 was transformed with the indicated *STE11* mutant gene carried by a single-copy plasmid that is expressed from the *STE11* promoter: vec, vector; WT, wild-type; DDD, S281D/S285D/T286D. Cells were incubated with (+) or without (−) 1 M NaCl for 5 min.

E–H   Analyses of Hog1 phosphorylation by Phos-tag band-shift assay. Yeast strains (E) KY603-3; (F) TM142; (G) TM257; and (H) FP54 were stimulated with the indicated concentrations of NaCl for 5 min.

I   Comparison of the NaCl dose–responses of Hog1 activation by various strains. Phos-tag band-shift assays shown in (C and E–H) were independently repeated three times, and average values were plotted.

Data information: (C and E–H) Representative results from three independent experiments. (I) Error bars are SEM (*n* = 3).
Source data are available online for this figure.

## The downstream osmosensor acts at the step of Hog1 phosphorylation by Pbs2

Next, we identified the signaling step in the HOG pathway at which the predicted downstream osmosensor functions. It has been reported that Hog1 can, when strongly overexpressed, auto-phosphorylate in the absence of Pbs2 (Maayan *et al*, 2012). However, Hog1 auto-phosphorylation could not account for our observations, because in our experiments Hog1 was not overexpressed, and the catalytically inactive Hog1-K52S/K53N (Alepuz *et al*, 2001) could be phosphorylated by osmostress in the absence of upstream osmosensors (Fig 2A, lanes 4–6). On the contrary, Pbs2 was required to phosphorylate Hog1 by the downstream osmosensor (Fig 2B, lanes 1–4). Furthermore, neither expression of Pbs2-S514A/T518A, which lacked activating phosphorylation sites, nor that of catalytically inactive Pbs2-K389M supported Hog1 phosphorylation by osmostress (Fig 2B, lanes 5–8), further indicating that active Pbs2 is necessary for Hog1 phosphorylation induced by the downstream osmosensor. In contrast, MAP3Ks were not required to phosphorylate Hog1 by the downstream osmosensor if the constitutively active Pbs2-S514D/T518D (Pbs2-DD) was present (Fig 2C, compare lanes 6 and 9). Phosphorylation of Hog1 in the presence of Pbs2-DD had a very similar dose–response to that in the presence of Ste11-Q301P (Fig 2D and E), indicating that Pbs2-DD phosphorylated Hog1 by the same mechanism. We thus concluded that osmostress enhances the Hog1 phosphorylation at the step of the Pbs2-Hog1 reaction (Fig 2F).

Generally speaking, two mechanisms are possible at the step of the Pbs2-Hog1 reaction to increase the Hog1 phosphorylation: (i) enhancement of Hog1 phosphorylation by Pbs2, and (ii) inhibition of Hog1 dephosphorylation by phosphatases. An excellent example of the latter mechanism is the activation of Hog1 by arsenite, which inhibits the major Hog1 phosphatases Ptp2 and Ptp3 (Lee & Levin, 2018). We thus examined if the osmotic enhancement of Hog1 phosphorylation at the step of the Pbs2-Hog1 reaction also involves the inhibition of these phosphatases. In a ΔS/O/H/M *ssk2/22Δ STE11-Q301P* strain, deletion of either *PTP2* or *PTP3* alone had essentially no effect (Fig EV2). As expected, deletion of both *PTP2* and *PTP3* together increased the Hog1 phosphorylation even without osmotic stress. More important, 5-min treatment of the *ptp2Δ ptp3Δ* strain further increased the extent of Hog1 phosphorylation. It should be noted that about 30% of the

unphosphorylated Hog1 present before NaCl addition was phosphorylated both in the *PTP2 PTP3* and *ptp2 ptp3* strains. Thus, osmotic enhancement of Hog1 phosphorylation must be attained by promotion of Hog1 phosphorylation by Pbs2, but not by inhibition of Hog1 dephosphorylation.

## The Hog1 L16 domain is necessary for the osmotic enhancement of the Pbs2-Hog1 reaction

To investigate the physiological role as well as the molecular mechanism of the osmotic enhancement of the Pbs2-Hog1 reaction, we tried to isolate Hog1 mutants that are unable to be enhanced. We based our screen for Hog1 mutants that cannot be osmotically enhanced on the expectation that such Hog1 mutants would not be phosphorylated at 1.0 M NaCl in the ΔS/O/H/M *ssk2/22Δ STE11-Q301P* strain. Furthermore, we thought that the C-terminal non-catalytic region of Hog1 was especially promising for the search of Hog1 mutants that cannot be osmotically enhanced, as this domain contains the highly conserved common docking (CD) domain that binds Pbs2 (Murakami *et al*, 2008) and the moderately conserved L16 domain (Fig 3A) that are known to modulate Hog1 activation (Maayan *et al*, 2012).

We constructed Hog1 expression plasmids that lacked various parts of the C-terminal non-catalytic region (Fig 3B). These constructs were individually introduced into the ΔS/O/H/M *ssk2/22Δ STE11-Q301P hog1Δ* strain, and their phosphorylation was assayed in the absence or presence of osmostress. There was no phosphorylation of the Hog1 deletion mutant that lacked most of the non-catalytic C-terminal domain (Δ[320–431]; Fig 3C, lanes 7–9). In contrast, the Hog1 deletion mutant that retained the L16 domain (Hog1Δ[355–431]) was phosphorylated in the presence of osmostress (Fig 3C, lanes 4–6). Finally, deletion of the L16 region alone (ΔL16 = Δ[320–350]) was sufficient to inhibit Hog1 phosphorylation, whereas *HOG1* mutation D307A/D310A that abrogated the binding capacity of the CD domain (Murakami *et al*, 2008) had little effect (Fig 3D and E). The ΔL16 mutation also suppressed Hog1 phosphorylation driven by Pbs2-DD (Fig 3F and G). We concluded from these results that the Hog1 L16 domain was required for osmotic enhancing of Hog1 phosphorylation.

Although Hog1-ΔL16 could not be phosphorylated by Pbs2-DD, it was phosphorylated in WT cells in which both the SLN1 and SHO1 branches are intact (Fig 3H), suggesting that normally activated

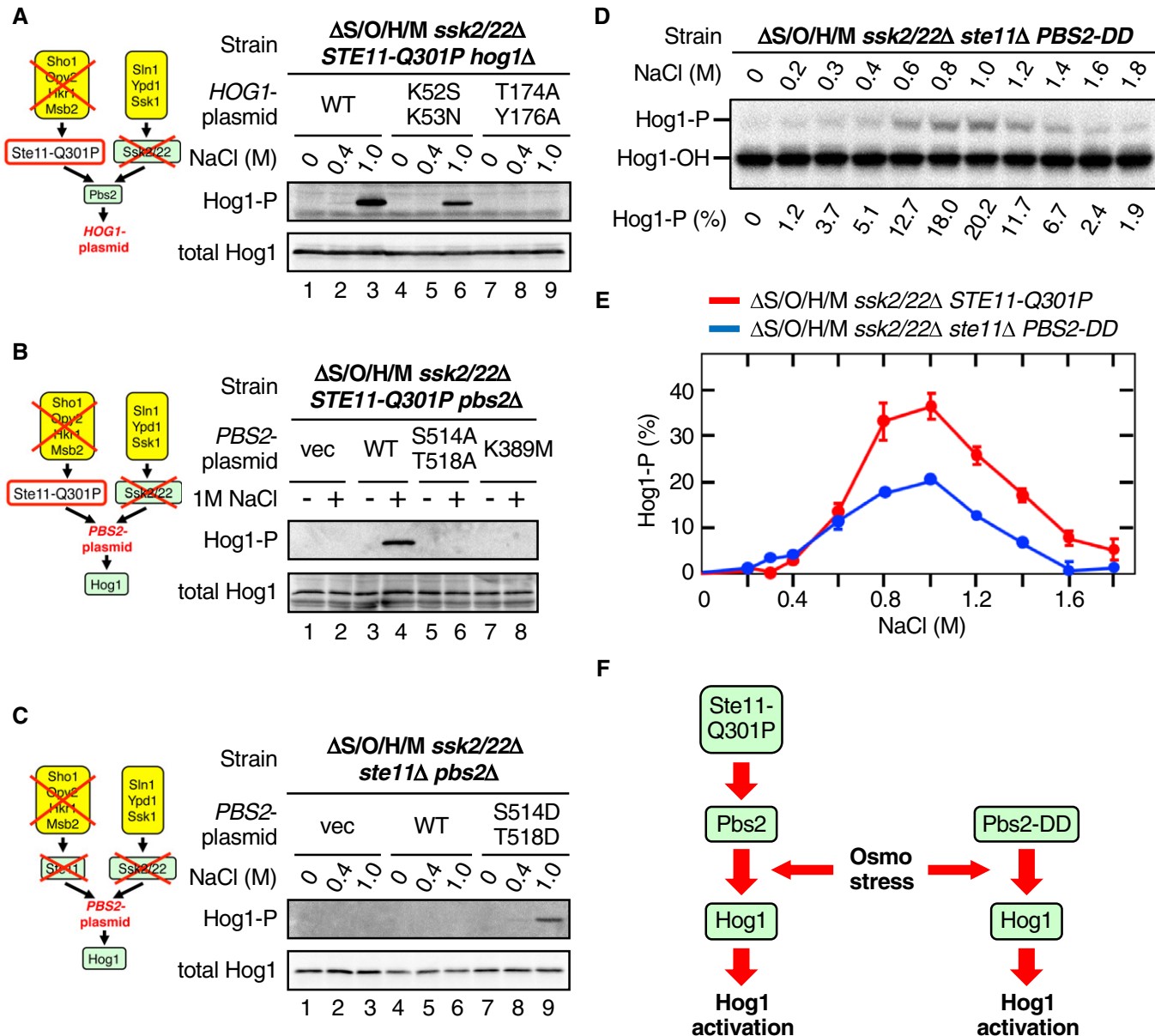

**Figure 2. Osmostress enhances the phosphorylation of Hog1 by Pbs2.**

A–C  Immunoblot analyses of Hog1 phosphorylation. Yeast strains (A) KT235; (B) KT209; and (C) KT234 were stimulated with the indicated concentrations of NaCl for 10 min, and phosphorylated Hog1 (Hog1-P) and total Hog1 in cell lysates were detected by immunoblotting. The relevant genotypes of the strains are indicated in the top row of each panel and are schematically shown in the diagrams at left. The second row from the top indicates the genes carried by a single-copy plasmid that are expressed from their own promoters. vec, vector; WT, wild-type.

D  Phos-tag band-shift analyses of Hog1 phosphorylation. The yeast strain KT234 carrying the single-copy expression plasmid YCplac22I'-Pbs2 S514D/T518D was stimulated with the indicated concentrations of NaCl for 5 min.

E  Comparison of the NaCl dose–responses of Hog1 activation by constitutively active Ste11-Q301P and constitutively active Pbs2-DD. Phos-tag band-shift assays shown in (D) and Fig 1E were independently repeated three times, and the average values were plotted.

F  A scheme illustrating the step in the Hog1 MAPK cascade at which osmostress acts to enhance Hog1 phosphorylation.

Data information: (E) Error bars are SEM ($n = 3$).
Source data are available online for this figure.

Pbs2 (by the upstream MAP3Ks) can phosphorylate Hog1 without the osmotic enhancement of the Pbs2-Hog1 reaction. If so, these results suggest that Pbs2-DD, though mimicking the phosphorylated Pbs2, has a significantly weaker activity than the normally phosphorylated Pbs2.

**Inhibition of the osmotic enhancement of the Pbs2-Hog1 reaction affects the SLN1 and SHO1 branches differently**

Next, we examined how inhibition of the osmotic enhancement by ΔL16 affected the Hog1 phosphorylation mediated by individual

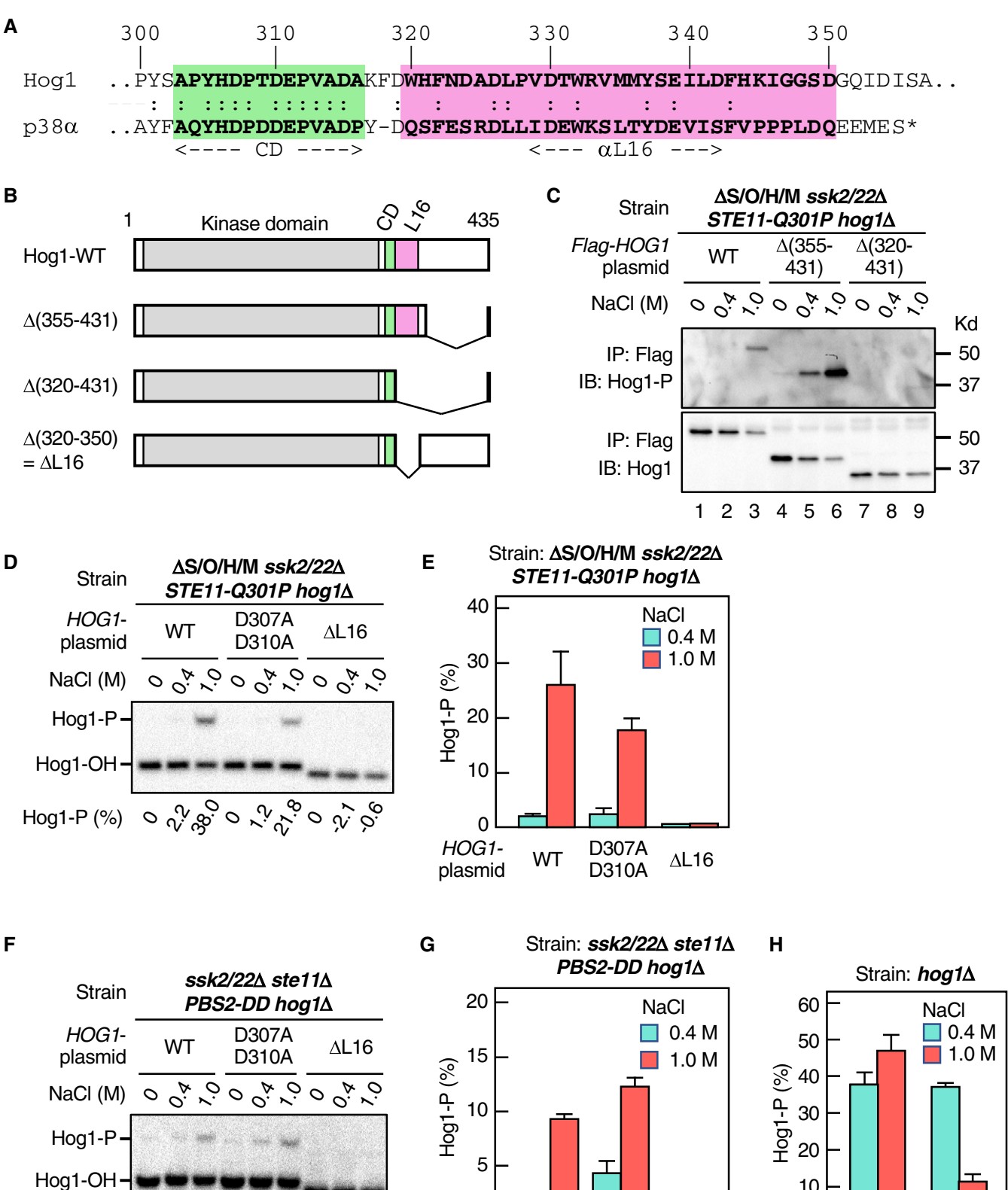

**Figure 3.**

**Figure 3. The Hog1 L16 domain is required for the osmotic enhancement of the Pbs2-Hog1 reaction.**

A   Alignment of the amino acid sequences of the CD (green) and L16 (pink) domains of yeast Hog1 and mammalian p38α. The alpha helix αL16 forms the core of the L16 domain (Wang *et al*, 1997).

B   Schematic diagrams of Hog1-WT and its deletion constructs used in this study.

C   Immunoblot analyses of Hog1 phosphorylation. The yeast strain KT235 was transformed with pRS416-FLAG-Hog1 (WT) or its indicated deletion derivatives. FLAG-Hog1 was immunoprecipitated (IP), and immunoblotted (IB) with anti-phospho-p38 (for Hog1-P; upper panel) or anti-FLAG (for total FLAG-Hog1; lower panel)

D–G   Phos-tag band-shift assay of Hog1 phosphorylation. Yeast strain (D and E) KT235 or (F and G) KT290 carrying the single-copy expression plasmid YCplac22I'-Pbs2 S514D/T518D was transformed with either pRS416-Hog1 (WT) or its indicated mutant derivatives and was treated with the indicated concentrations of NaCl for 5 min. (D) and (F) show typical results, and (E) and (G) summarize the averages of three independent experiments.

H   Phos-tag band-shift assay of Hog1 phosphorylation. The yeast strain FP4 was transformed with the single-copy expression plasmid pRS416-Hog1 (WT) or pRS416-Hog1-ΔL16 and was treated with the indicated concentrations of NaCl for 5 min. The averages of three independent experiments are shown.

Data information: (E, G, and H) Error bars are SEM (*n* = 3).
Source data are available online for this figure.

upstream osmosensing branches. For this purpose, we expressed Hog1-ΔL16 in a WT strain (with both the SLN1 and SHO1 branches), an *ste11Δ* strain (the SLN1 branch-only), or an *ssk2/22Δ* strain (the SHO1 branch-only) and measured Hog1-ΔL16 phosphorylation at various NaCl concentrations. In the WT strain (Fig 4A and B) and the SLN1 branch-only strain (Fig 4C and D), phosphorylation of Hog1-ΔL16 was strongly reduced at higher NaCl concentrations (> 1.0 M) compared to that of Hog1-WT, whereas it was comparable to that of Hog1-WT at lower NaCl concentrations (< 0.4 M). In contrast, in the SHO1 branch-only strain, phosphorylation of Hog1-ΔL16 was strongly reduced at both lower and higher NaCl concentrations compared to that of Hog1-WT (Fig 4E and F). This observation was quite puzzling, as the extent of Hog1-WT phosphorylation at the lower NaCl concentration range (0.2–0.6 M NaCl) is about the same for the three strains examined. To explain this difference between the SLN1 and SHO1 branches, we hypothesized that there might be a qualitative difference in the state of activated Pbs2 depending on whether it was activated by Ssk2/22 or by Ste11.

**Detection of Pbs2 phosphorylation at S514 and T518**

To investigate the above hypothesis, we examined the status of the Pbs2 phosphorylation following its activation by Ssk2/22 or by Ste11. Ssk2/22 and Ste11 are thought to activate Pbs2 by phosphorylating the conserved Ser-514 and Thr-518 (S514 and T518) in the activation loop (Fig EV3A). Indeed, mutation of both residues to Ala completely inhibits Pbs2 activation following osmostress (Maeda *et al*, 1995). However, to our best knowledge, it has not been demonstrated that these two sites are actually phosphorylated by Ste11 and/or Ssk2/22.

First, we examined Pbs2 phosphorylation using a Phos-tag band-shift assay. For that purpose, we constructed a C-terminally HA-tagged Pbs2 (Pbs2-HA) and expressed it from the *PBS2* promoter carried on a single-copy plasmid, to ensure that its intracellular concentration would be similar to its native expression level. When Pbs2-HA was expressed in WT (i.e., except the necessary *pbs2Δ* mutation) cells, a clear band shift of Pbs2 was observed in the Phos-tag assay upon application of osmostress (Fig EV3B, lanes 1–3). This band shift was dependent on the Pbs2 activating phosphorylation sites, as no band shift of Pbs2-HA-S514A/T518A was observed (Fig EV3B, lanes 4–6). The band shift was not due to Pbs2 auto-phosphorylation or to a retrograde phosphorylation by activated Hog1, as band shifts were observed for a kinase-dead Pbs2 (Pbs2-

HA-K389M) and for Pbs2-HA in a *hog1Δ* mutant (Fig EV3B, lanes 7–12). To test if Pbs2 phosphorylation at both S514 and T518 contributed to this band shift, we constructed the single Ala substitution mutants S514A and T518A. A band shift of Pbs2-HA-T518A was observed upon osmostress, and its extent was indistinguishable from that of Pbs2-HA-WT, indicating that phosphorylation at S514 alone was sufficient to induce the observed band shift (Fig EV3C, lanes 7–9). In contrast, no band shift was observed for Pbs2-HA-S514A (Fig EV3C, lanes 4–6), even though T518 phosphorylation did occur in this mutant (see below). Thus, the Phos-tag band-shift assay specifically detects the phosphorylation of S514, irrespective of the phosphorylation status of T518.

Because the Phos-tag band-shift assay cannot be used to detect the phosphorylation at T518, we generated a polyclonal rabbit antibody against phosphorylated T518. Immunoblotting by anti-phospho-T518 of the same samples used in the above band-shift assays indicated that a positive signal was dependent on the Pbs2 activating phosphorylation sites (Fig EV3D, lanes 1–6) and that this antibody signal specifically reflected T518 phosphorylation (Fig EV3E). Thus, by using the Phos-tag band-shift assay and the anti-phospho-T518 immunoblotting assay, it was possible to monitor the phosphorylation of Pbs2 at S514 and T518 separately.

**Ste11 and Ssk2/Ssk22 differentially phosphorylate Pbs2**

We first examined S514 phosphorylation using the band-shift method. When either WT cells or SLN1 branch-only cells (*ste11Δ*) were osmostressed, S514 phosphorylation was strongly induced (Fig 5A, lanes 1–6). In contrast, no S514 phosphorylation was detected in the SHO1 branch-only cells (*ssk2/22Δ*; Fig 5A, lanes 7–9). As expected, no S514 phosphorylation was observed in mutant cells that lacked the three MAP3Ks (*ste11Δ ssk2/22Δ*; Fig 5A, lanes 10–12). Mass spectrometric analyses of phosphopeptides from 0.6 M NaCl-treated cells also showed that peptides containing phospho-S514 could be detected in the SLN1 branch-only cells, but not in the SHO1 branch-only cells (Fig EV3F). These results suggested that S514 was phosphorylated only by Ssk2/22.

We next examined the T518 phosphorylation using the anti-phospho-T518 antibody. Osmostress (0.6 and 1 M NaCl) induced T518 phosphorylation in the WT cells, the SLN1 branch-only cells (*ste11Δ*), and the SHO1 branch-only cells (*ssk2/22Δ*; Fig 5B, lanes 1–9). As expected, no T518 phosphorylation was observed in the *ste11Δ ssk2/22Δ* cells (Fig 5B, lanes 10–12; note that the faint band in these lanes is slightly smaller in size than the phosphorylated

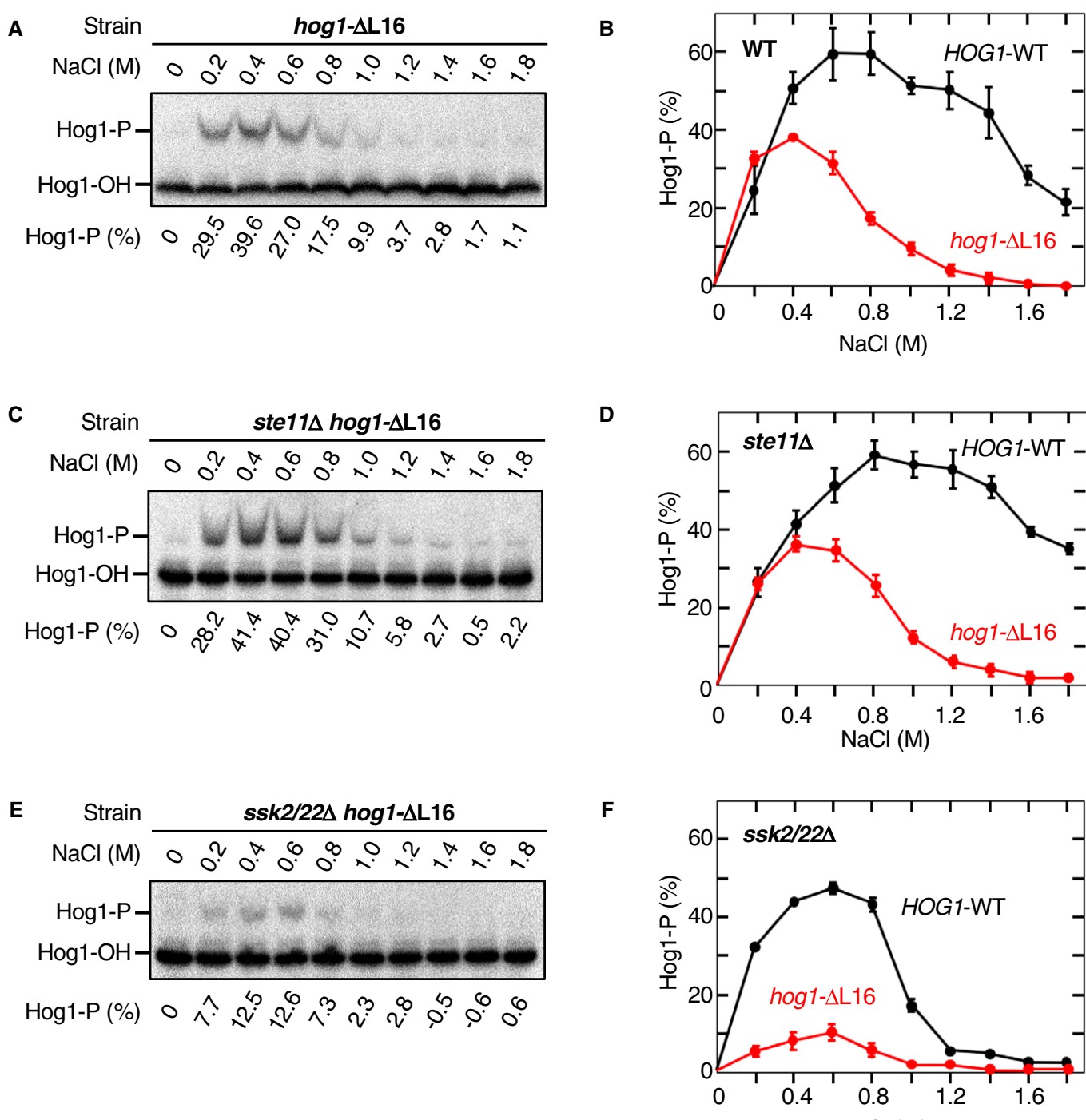

**Figure 4. Inhibition of the osmotic enhancement of the Pbs2-Hog1 reaction affects the SLN1 and SHO1 branches differently.**

A    Phos-tag band-shift analyses of osmostress-induced Hog1 phosphorylation. The yeast strain FP4 (*hog1Δ*) was transformed with the single-copy expression plasmid pRS416-Hog1-ΔL16 and was stimulated with the indicated concentrations of NaCl for 5 min.

B    Averages of three independent experiments from A were plotted. Results for Hog1-WT (from Fig 1I) are included for comparison.

C, D    Same as in (A and B), except that the yeast strain KT259 (*ste11Δ hog1Δ*) was used.

E, F    Same as in (A and B), except that the yeast strain KY523 (*ssk2/22Δ hog1Δ*) was used.

Data information: (A, C, and E) Representative results from three independent experiments. (B, D, and F) Error bars are SEM (*n* = 3).
Source data are available online for this figure.

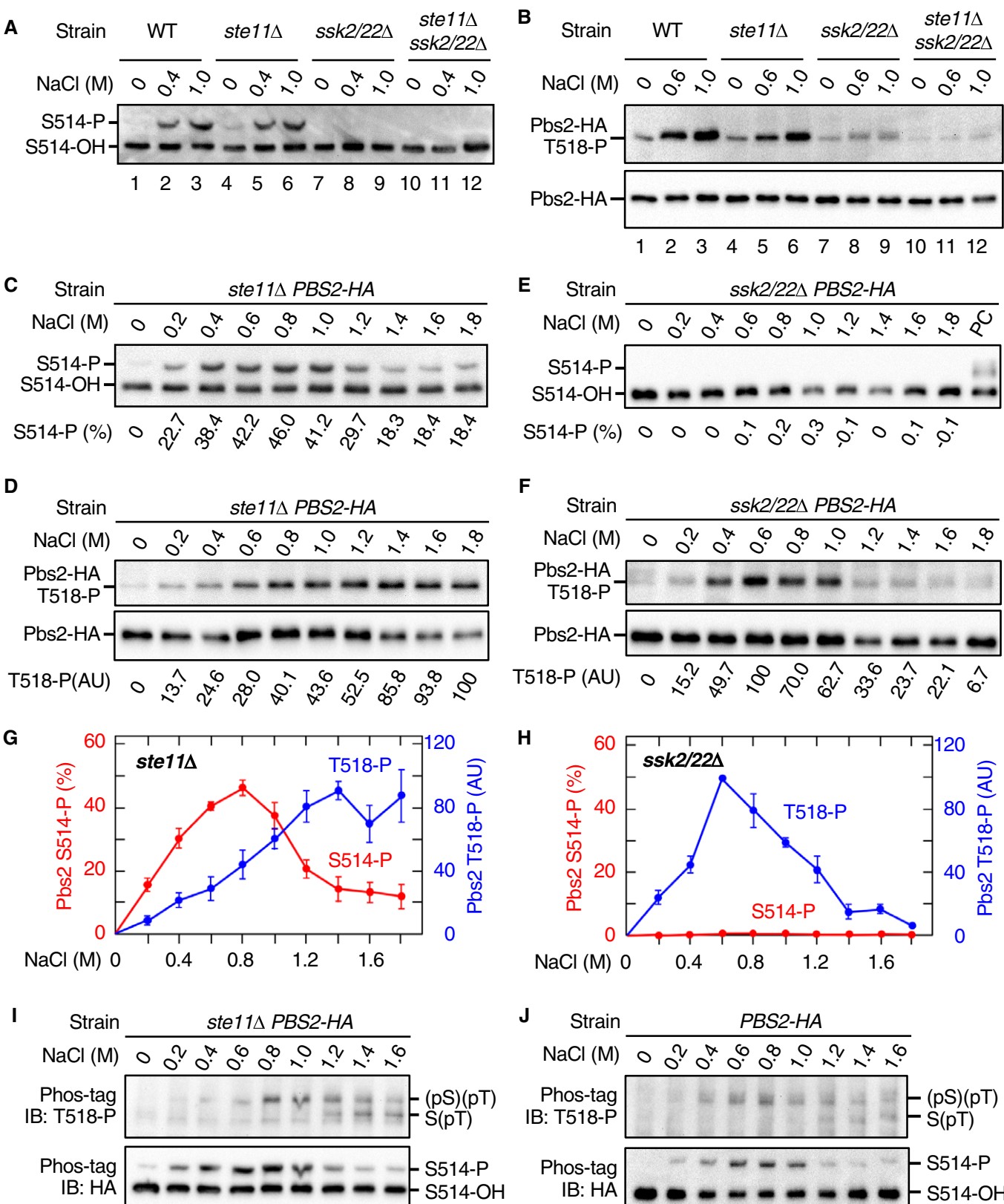

**Figure 5.**

◀

**Figure 5. Ste11 and Ssk2/Ssk22 differentially phosphorylate Pbs2.**

A, B  Detection of Pbs2 phosphorylation at S514 and T518. The yeast strains KT003 (*pbs2Δ*), KT005 (*ste11Δ pbs2Δ*), TM280 (*ssk2/22Δ pbs2Δ*), and KT043 (*ste11Δ ssk2/22Δ pbs2Δ*) were transformed with YCplac22I'-Pbs2-HA and were treated with the indicated concentrations of NaCl for 5 min. Pbs2-HA was immunoprecipitated from cell extract, and phosphorylated Pbs2 was analyzed by (A) Phos-tag band-shift assay or (B) anti-phospho-T518 immunoblotting. In (A), the positions of Pbs2-HA phosphorylated (S514-P) and unphosphorylated (S514-OH) at S514 are indicated.

C–F  NaCl dose–response analyses of Pbs2 phosphorylation. (C and D) KT005 (*ste11Δ pbs2Δ*) and (E and F) TM280 (*ssk2/22Δ pbs2Δ*) were transformed with YCplac22I'-Pbs2-HA and were treated with the indicated concentrations of NaCl for 5 min. (C and E) S514 phosphorylation was analyzed using the Phos-tag band-shift assay. PC; positive control. (D and F) T518 phosphorylation was analyzed using anti-phospho-T518 immunoblotting.

G, H  Average values of three independent experiments from (C and D) and (E and F), respectively, were plotted. AU, arbitrary unit.

I  Detection of di-phosphorylated Pbs2. KT005 (*ste11Δ pbs2Δ*) transformed with YCplac22I'-Pbs2-HA was treated with the indicated concentrations of NaCl for 5 min. Pbs2-HA was immunoprecipitated from cell extracts and subjected to Phos-tag SDS–PAGE. Blots of these gels were probed with (upper panel) anti-phospho-T518 or (lower panel) anti-HA.

J  Same as in (I), except that the yeast strain KT003 (*pbs2Δ*) was used.

Data information: (C–F) Representative results from three independent experiments. (G and H) Error bars are SEM (*n* = 3).
Source data are available online for this figure.

T518 and probably is a non-specific band). These results indicated that T518 could be phosphorylated either by Ssk2/22 or by Ste11.

To examine the difference between the two branches in more detail, we performed a NaCl dose–response analysis of Pbs2 phosphorylation at S514 and T518. In the SLN1 branch-only cells (*ste11Δ*), strong S514 phosphorylation was detected between 0.4 and 1.2 M NaCl (Fig 5C) with relatively weak phosphorylation detected at 0.2 M and above 1.4 M NaCl. In the same cells, T518 phosphorylation gradually increased as the NaCl concentration increased, until it reached a plateau at 1.4 M NaCl (Fig 5D). In the SHO1 branch-only cells (*ssk2/22Δ*), no S514 phosphorylation was detected at any NaCl concentration between 0.2 M and 1.8 M (Fig 5E), whereas strong T518 phosphorylation was observed between 0.4 and 1.0 M NaCl (Fig 5F). The NaCl dose–responses of phosphorylation at S514 and T518 are plotted in Fig 5G (SLN1 branch) and Fig 5H (SHO1 branch). These data confirmed that S514 was only phosphorylated by Ssk2/22, whereas T518 was phosphorylated by both Ssk2/22 and Ste11.

Although the preceding analyses proved that both S514 and T518 were phosphorylated in the SLN1 branch-only cells, they did not provide any direct evidence that di-phosphorylated Pbs2 [Pbs2-(pS)(pT)] was generated in the same cells. To examine whether Pbs2-(pS)(pT) was generated or not, we combined the two assay methods together. Briefly, the SLN1 branch-only cells (*ste11Δ PBS2-HA*) were osmostressed, and immunoprecipitated Pbs2-HA was separated by Phos-tag SDS–PAGE according to its S514 phosphorylation statuses. The blot was subsequently probed with anti-phospho-T518 antibody (Fig 5I, upper blot) or with anti-HA antibody (lower blot). In the upper blot, the upper bands reflect di-phosphorylated Pbs2-(pS)(pT) whereas the lower bands reflect mono-phosphorylated Pbs2-S(pT). Clearly, di-phosphorylated Pbs2-(pS)(pT) was generated in the SLN1 branch-only cells. Analysis of the WT cells (with both SLN1 and SHO1 branches) by the same method gave a result essentially identical to that of the SLN1 branch-only cells (Fig 5J), indicating that there is no significant synergism and/or interference between the SLN1 and the SHO1 branches.

**Mono-phosphorylated Pbs2 can phosphorylate Hog1 only when the Pbs2-Hog1 reaction is osmotically enhanced**

In Fig 4, we showed that phosphorylation of Hog1 by the SLN1 branch is much more resistant to the inhibition of osmotic

enhancement than that by the SHO1 branch. In Fig 5, we showed that Pbs2 is phosphorylated both at S514 and T518 by the SLN1 branch, whereas Pbs2 is phosphorylated only at T518 by the SHO1 branch. Thus, the enhancement-independent Hog1 activation by the SLN1 branch might be due to either the presence of S514-phosphorylated Pbs2 or that of di-phosphorylated Pbs2. To answer this question, we used the observation that Pbs2-S514A and Pbs2-T518A (Fig 6A) could be mono-phosphorylated at T518 and S514, respectively (see Fig EV3C and E). Expression of Pbs2-HA-S514A or Pbs2-HA-T518A in a SLN1 branch-only strain (*ste11Δ*) supported Hog1 phosphorylation upon osmostress, indicating that mono-phosphorylated Pbs2 could phosphorylate Hog1 (Fig 6B, lanes 3–6). Pbs2-WT phosphorylated Hog1 more efficiently than either S514A or T518A did, likely because Pbs2-WT could be di-phosphorylated in the *ste11Δ* strain. The double mutant Pbs2-HA-S514A/T518A could not phosphorylate Hog1, indicating at least one of the phosphorylation events in Pbs2 is necessary for Hog1 phosphorylation (Fig 6B, lanes 7–8).

Next, to test if either of the mono-phosphorylated Pbs2 (i.e., S514-P and T518-P) could phosphorylate Hog1 without the osmotic enhancement, we expressed Pbs2-HA-S514A or Pbs2-HA-T518A together with either Hog1-WT or the Hog1-ΔL16 (which is insensitive to osmotic enhancement), in an SLN1 branch-only strain (*ste11Δ pbs2Δ hog1Δ*), and Hog1 phosphorylation was examined upon osmostress. Both Pbs2-HA-S514A and Pbs2-HA-T518A could phosphorylate Hog1-WT between 0.4 M and 1.2 M NaCl (Fig 6C and D), but induced little or no phosphorylation of Hog1-ΔL16 at any NaCl concentration (Fig 6E and F). The averages of three independent experiments are plotted in Fig 6G and H, respectively. These results provide clear experimental evidence that both mono-phosphorylated Pbs2 species, namely Pbs2-S(pT) and Pbs2-(pS)T, can phosphorylate Hog1 only when the Pbs2-Hog1 reaction is enhanced by osmostress (schematically summarized in Fig 6I). We thus deduce that enhancement-independent Hog1 phosphorylation must be mainly carried out by di-phosphorylated Pbs2-(pS)(pT).

**The Hog1-N149H/D162G mutant is constitutively enhanced**

The finding that the Hog1-ΔL16 is an enhancement-defective mutant prompted us to screen for the reverse type of Hog1 mutants, namely constitutively enhanced Hog1 mutants. The expected hallmark of such a mutant was its ability to be

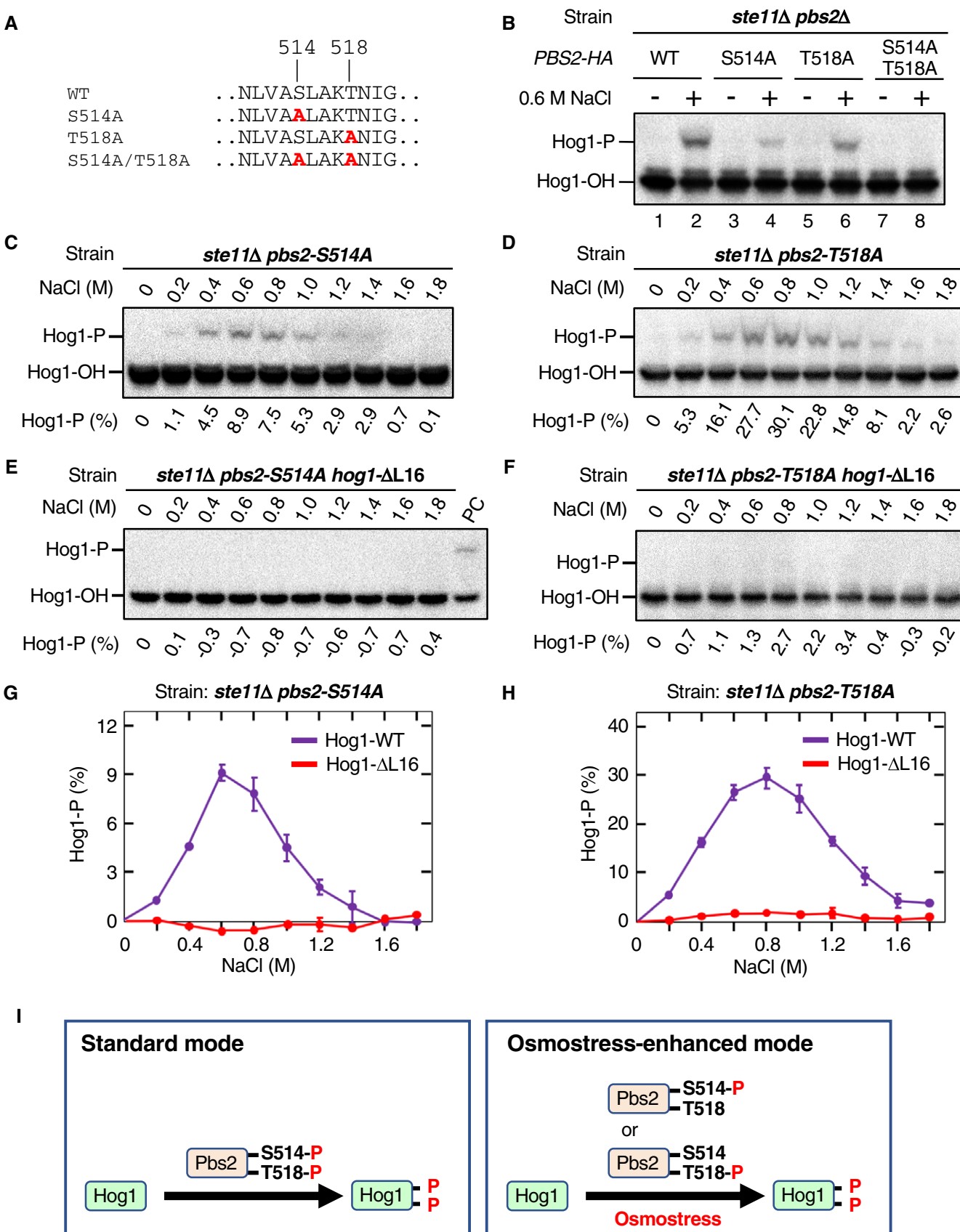

**Figure 6.**

**Figure 6.** Mono-phosphorylated Pbs2 can phosphorylate Hog1 only when the Pbs2-Hog1 reaction is enhanced.

A  The Pbs2 single and double Ala substitution mutations used in this study.
B  Phos-tag band-shift analyses of osmostress-induced Hog1 phosphorylation. KT005 (*ste11Δ pbs2Δ*) was transformed with YCplac22I'-Pbs2-HA (WT) or its indicated mutant derivatives. Cells were stimulated with (+) or without (−) 0.6 M NaCl for 5 min.
C–F  Phos-tag band-shift analyses of Hog1 phosphorylation. KT291 (*ste11Δ pbs2Δ hog1Δ*) was transformed with (C and D) pRS416-Hog1 or (E and F) pRS416-Hog1-ΔL16, together with (C and E) YCplac22I'-Pbs2-S514A-HA or (D and F) YCplac22I'-Pbs2-T518A-HA. Cells were stimulated with the indicated concentrations of NaCl for 5 min.
G, H  Average values of three independent experiments from (C and E) and (D and F), respectively, are plotted.
I  A scheme illustrating the osmotic enhancement of the Pbs2-Hog1 reaction in activating Hog1 by mono-phosphorylated Pbs2.

Data information: (C–F) Representative results from three independent experiments. (G and H) Error bars are SEM (*n* = 3).
Source data are available online for this figure.

phosphorylated by mono-phosphorylated Pbs2 in the absence of osmostress (i.e., it does not require osmotic enhancement). In practice, we screened for a Hog1 mutant that could be phosphorylated by Pbs2 if the cells expressed Ste11-Q301P (which mimics the activation of the SHO1 branch, and likely produces only mono-phosphorylated Pbs2), in the absence of osmostress and upstream osmosensors. For details of mutant isolation, see Materials and Methods. To date, only one mutant with the required characteristic, *HOG1-N149H/D162G*, has been isolated.

We first examined the phosphorylation status of Hog1-N149H/D162G using the Phos-tag band-shift assay. In the ΔS/O/H/M *ssk2/22Δ STE11-Q301P* strain, Hog1-N149H/D162G was highly phosphorylated in the absence of osmostress, whereas there was no detectable phosphorylation of Hog1-WT (Fig 7A, lanes 1–3 and 10–12, each set is an independent triplicate). Mutants with only one mutation, Hog1-N149H and Hog1-D162G, were not phosphorylated under the same conditions (Fig 7A, lanes 4–9), indicating that both N149H and D162G were necessary to generate a constitutively enhanced Hog1 mutant. Hog1-N149H/D162G was also phosphorylated in ΔS/O/H/M *ssk2/22Δ PBS2-DD* cells in the absence of osmostress (Fig 7B, blue). Since active Ste11 generates only the mono-phosphorylated Pbs2-S(pT), which cannot phosphorylate Hog1 unless the Pbs2-Hog1 reaction is osmotically enhanced, these results are consistent with the interpretation that Hog1-N149H/D162G mimicked the osmotically enhanced state.

It is previously reported that catalytically inactive Hog1 is constitutively phosphorylated because of the lack of negative feedback (Wurgler-Murphy *et al*, 1997). Thus, if Hog1-N149H/D162G was catalytically compromised, the results in Fig 7A and B might be explained. However, Hog1-N149H/D162G was catalytically active, because it could support a strong induction of the Hog1-dependent reporter gene *8xCRE-lacZ* (Fig 7C). Thus, enhanced phosphorylation of Hog1-N149H/D162G was not due to a lack of negative feedback.

In the same mutant screening described above, we also isolated a Hog1 mutant that is constitutively active even in the absence of Pbs2. This mutant, Hog1-F318S/H344L, contains the previously reported mutation F318S that allows Hog1 auto-phosphorylation in the absence of Pbs2 at high osmolarities (Maayan *et al*, 2012). We hypothesized that Hog1-F318S has in its unphosphorylated state a slight catalytic activity toward its own activating phosphorylation sites, and osmostress enhances the auto-phosphorylating reaction in a similar manner as osmostress enhances the Pbs2-Hog1 reaction. If the Hog1-N149H/D162G mutation mimics the effect of osmostress, then combining it with the F318S mutant should allow Hog1 auto-activation in the absence of osmostress in the *pbs2Δ* background. We thus examined the expression of the Hog1 activity reporter

*8xCRE-LacZ* in a *pbs2Δ* strain that expressed Hog1-N149H/D162G/F318S/H344L. In the absence of osmostress, neither Hog1-WT nor Hog1-N149H/D162G induced the Hog1 reporter, whereas Hog1-F318S/H344L induced very weak, but detectable expression of the reporter (Fig 7D). The combination of both mutations, however, led to a very powerful Hog1 activation, supporting our hypothesis that Hog1-N149H/D162G mimics the osmostressed condition. Even more important, this result indicated that the osmotic enhancement of Hog1 phosphorylation was not specific to Pbs2. We conclude that the Hog1 activating phosphorylation sites become more accessible to phosphorylating kinases (either Pbs2 or Hog1) in the presence of osmostress or by the N149H/D162G mutation.

### The lack of osmotic enhancement prevents non-specific activation of Hog1 in the absence of osmostress

A possible function of the osmotic enhancement is to reduce basal activation of Hog1 by weakly activated Pbs2 in the absence of osmostress. To test this idea, we determined the phosphorylation statuses of Hog1-WT and Hog1-N149H/D162G in cells that had defects in one or both upper branches of the HOG pathway, in the absence of osmostress. When Hog1-WT was expressed in any of these cells, it was barely phosphorylated in the absence of osmostress (Fig 7E, green). In contrast, Hog1-N149H/D162G was strongly phosphorylated in the WT cells (Fig 7E, blue), indicating that a substantial level of latent Pbs2 activity (most likely in the form of mono-phosphorylated Pbs2) was present even in the absence of osmostress. This finding further indicates that there must be relatively high basal activities of MAP3Ks present in the absence of osmostress. Inactivation of the SLN1 branch (*ssk2/22Δ*) had strongly reduced the phosphorylation of Hog1-N149H/D162G, indicating that the basal activity in the WT cells came mostly from the SLN1 branch, as previously reported (Macia *et al*, 2009). In contrast, inactivation of the SHO1 branch alone (*ste11Δ*) had almost no effect on the phosphorylation of Hog1-N149H/D162G. However, the slight activation of Hog1-N149H/D162G in the *ssk2/22Δ* cells was completely abolished by additional mutations in the components of the SHO1 branch (*sho1Δ*, *opy2Δ*, and *hkr1Δ msb2Δ*; Fig 7F), indicating that the SHO1 branch also had a basal activity in the absence of osmostress, although it is much weaker than that of the SLN1 branch.

An important inference from these findings is that both the SLN1 and the SHO1 branches have basal activities, but that the lack of the osmotic enhancement of the Pbs2-Hog1 reaction prevents Hog1 activation in the absence of osmostress (Fig 7G).

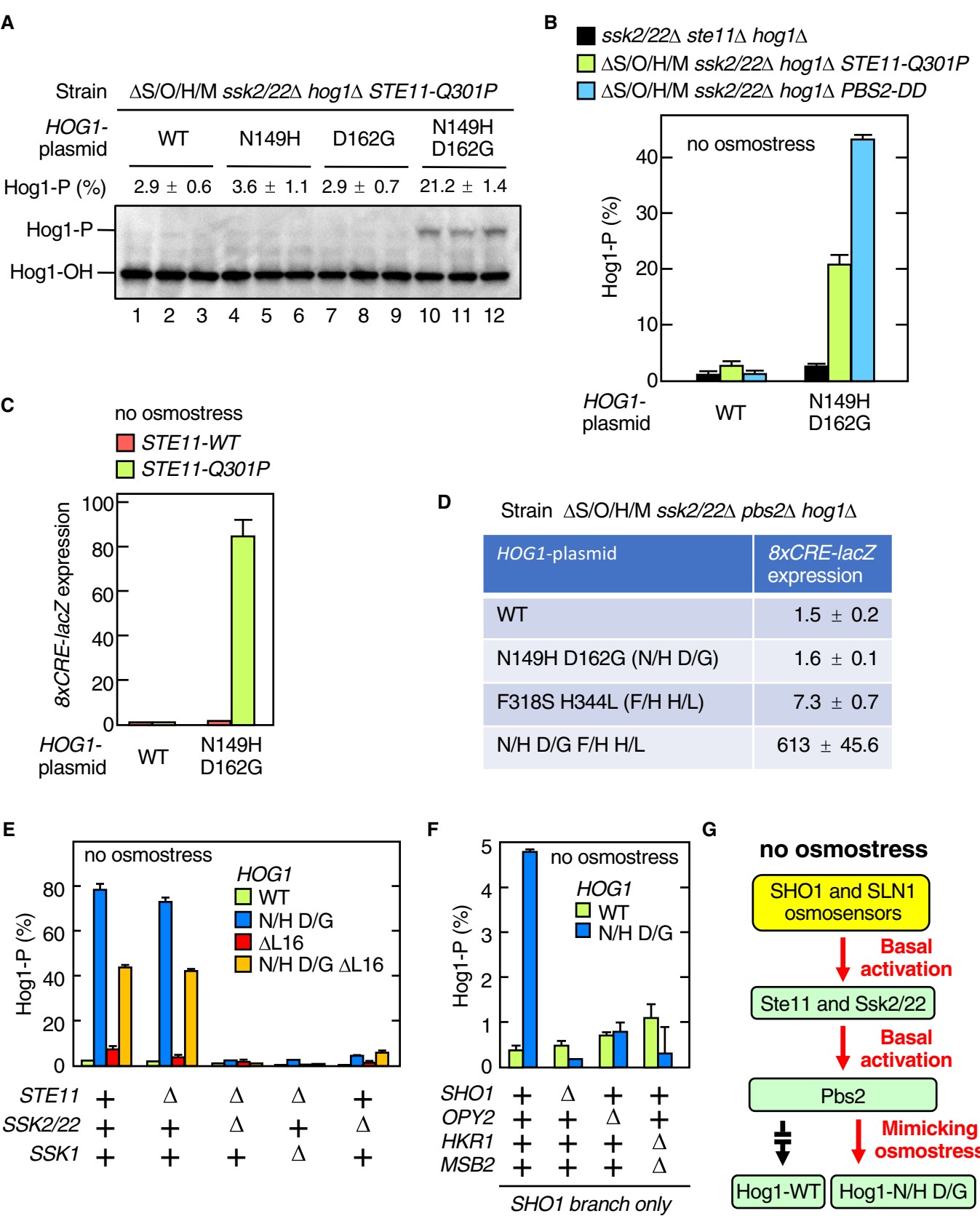

**Figure 7.**

◀

**Figure 7. The Hog1-N149H/D162G mutant constitutively enhanced the Pbs2-Hog1 reaction.**

A Phos-tag band-shift assay of Hog1 phosphorylation. KT235 (ΔS/O/H/M *ssk2/22Δ hog1Δ STE11-Q301P*) was transformed with pRS416-Hog1 (WT) or its indicated mutant derivatives. Cell extracts were prepared from fresh cultures without applying osmostress. For each *HOG1* mutant plasmid, three independent cultures were assayed.

B Same as in (A), except that the yeast strains KT260 (*ssk2/22Δ ste11Δ hog1Δ*), KT235 (ΔS/O/H/M *ssk2/22Δ hog1Δ STE11-Q301P*), and KT248 (ΔS/O/H/M *ssk2/22Δ hog1Δ pbs2Δ*) carrying YCplac22I'-Pbs2-DD were used.

C KT250 (ΔS/O/H/M *ssk2/22Δ hog1Δ STE11-WT*) and KT235 (ΔS/O/H/M *ssk2/22Δ hog1Δ STE11-Q301P*) were transformed with pRS416-Hog1 (WT) or pRS416-Hog1-N149H D162G (N149H D162G) together with pRS414-8xCRE-lacZ. Expression of the Hog1 reporter gene *8xCRE-lacZ* in the absence of osmostress was assayed.

D KT248 (ΔS/O/H/M *ssk2/22Δ hog1Δ pbs2Δ*) was transformed with pRS416-Hog1 (WT) or its indicated mutant derivatives together with pRS414-8xCRE-lacZ. Expression of the Hog1 reporter gene *8xCRE-lacZ* in the absence of osmostress was assayed.

E, F Phos-tag band-shift assay of Hog1 phosphorylation. Yeast strains of the indicated genotypes (shown below the graph) were transformed with pRS416-Hog1 (WT) or its indicated mutant derivatives (shown inside the graph). Strain used were as follows: (E) FP4, KT259, KT260, KT292, and KY523; and (F) KY523, KT293, KT294, and KT295. N/H, N149H; D/G, D162G.

G A schematic model showing that the lack of osmotic enhancement of the Pbs2-Hog1 reaction prevents the basal Hog1 activation.

Data information: (A and B, and D–F) Error bars are SEM (*n* = 3). (C) Error bars are SEM (*n* = 4).
Source data are available online for this figure.

## The lack of osmotic enhancement prevents pheromone-to-Hog1 crosstalk in the absence of osmostress

The mating pheromones, which activate Ste11, do not activate Hog1. This absence of pheromone-to-Hog1 crosstalk is commonly attributed to the action of scaffold proteins that insulate pheromone-activated Ste11 from interacting with Pbs2 (Posas & Saito, 1997; Harris *et al*, 2001). However, our above results would suggest that, even if pheromone-activated Ste11 interacted with Pbs2, it would only phosphorylate Pbs2-T518. Since mono-phosphorylated Pbs2 cannot phosphorylate Hog1 unless the Pbs2-Hog1 reaction is osmotically enhanced, pheromone treatment in the absence of osmostress should therefore not activate Hog1.

To test the hypothesis that pheromones can activate Hog1 if there is osmostress, we used yeast strains that are defective in both the SHO1 and SLN1 branches (*hkr1Δ msb2Δ ssk2/22Δ* and ΔS/O/H/M *ssk2/22Δ*). In the *hkr1Δ msb2Δ ssk2/22Δ* strain, the pheromone pathway is intact, and, as expected, addition of the mating pheromone (α-factor) induced phosphorylation of Fus3 and Kss1 (Fig 8A), indirectly indicating that Ste11 was activated. However, no Hog1 phosphorylation was induced in this strain by the α-factor in the absence of osmostress (Fig 8B, —NaCl). Stimulation by osmostress (0.8 M NaCl) alone induced only a weak Hog1 phosphorylation. In contrast, when the same strain was treated simultaneously with the α-factor and osmostress, Hog1 phosphorylation increased in an α-factor concentration-dependent manner (Fig 8B, 0.8 M NaCl; summarized in Fig 8C). Essentially identical results are obtained with the ΔS/O/H/M *ssk2/22Δ* strain, excluding the possibility that the Sho1 and Opy2 proteins are somehow involved in the pheromone-induced Hog1 activation (Fig 8D).

According to the above hypothesis, pheromones should activate the constitutively enhanced Hog1-N149H/D162G even in the absence of osmostress. In fact, in the *hkr1Δ msb2Δ ssk2/22Δ* strain, Hog1-N149H/D162G was phosphorylated by pheromone treatment in a time-dependent manner, whereas Hog1-WT was not phosphorylated at all (Fig 8E).

Thus, pheromone stimulation could induce Hog1 phosphorylation if the Pbs2-Hog1 reaction was enhanced either osmotically or mutationally. In the absence of osmostress, the Pbs2-Hog1 reaction is suppressed, and fortuitous Hog1 activation by the mating pheromones will be prevented (schematically shown in Fig 8F).

## Mammalian p38 MAPK might also utilize a similar enhancement mechanism

The mammalian p38 MAPK is structurally very similar to Hog1 and is also activated by osmostress (Han *et al*, 1994). To examine if p38 utilizes a similar enhancement mechanism to that of Hog1, we constructed and analyzed the p38α-N155H/D168G mutant that is equivalent to Hog1-N149H/D162G. A FRET-based p38 activity probe (Tomida *et al*, 2015) showed that p38α-N155H/D168G has substantially higher kinase activity than p38α-WT in unstimulated human cells (Fig 8G and H). This result suggests that p38-N155H/D168G is constitutively enhanced and is activated by basal MAP2K activities. Thus, it seems possible that osmotic activation of p38 requires both osmotic stimulation of the upstream osmosensors and osmotic enhancement of the p38 phosphorylation.

## Time courses of the Hog1 phosphorylation at various osmolarities

To characterize the osmotic enhancement of the Pbs2-Hog1 reaction in more detail, we studied the temporal pattern of Hog1 phosphorylation in ΔS/O/H/M *ssk2/22Δ STE11-Q301P* and compared it with those of other strains. An example of time-course experiment for osmostressed ΔS/O/H/M *ssk2/22Δ STE11-Q301P* cells is shown in Fig 9A, and their time courses at various NaCl concentrations are compiled in Fig 9B. The temporal patterns of Hog1 phosphorylation in this strain can be divided into two modes depending on the intensity of osmostress. At moderately high osmolarities (0.4–1.0 M NaCl), Hog1 phosphorylation rose rapidly, reaching a peak activity that positively correlated with the strength of osmostress, followed by a rapid decline (the rapid-and-transient mode). At severely high osmolarities (> 1.2 M NaCl), Hog1 phosphorylation was initially weak, but it slowly increased in later time points (the delayed-activation mode).

This biphasic activation kinetics is not unique to the downstream osmosensor-mediated Hog1 phosphorylation. The SHO1 branch-only strain (*ssk2/22Δ*) also exhibited the rapid-and-transient mode (0.2–0.6 M NaCl) and the delayed-activation mode (> 1.0 M NaCl), but it was also possible to recognize the third, prolonged-activation mode where maximal Hog1 phosphorylation was maintained for a duration (0.8 M NaCl; Fig 9C). It should be noted that the transition

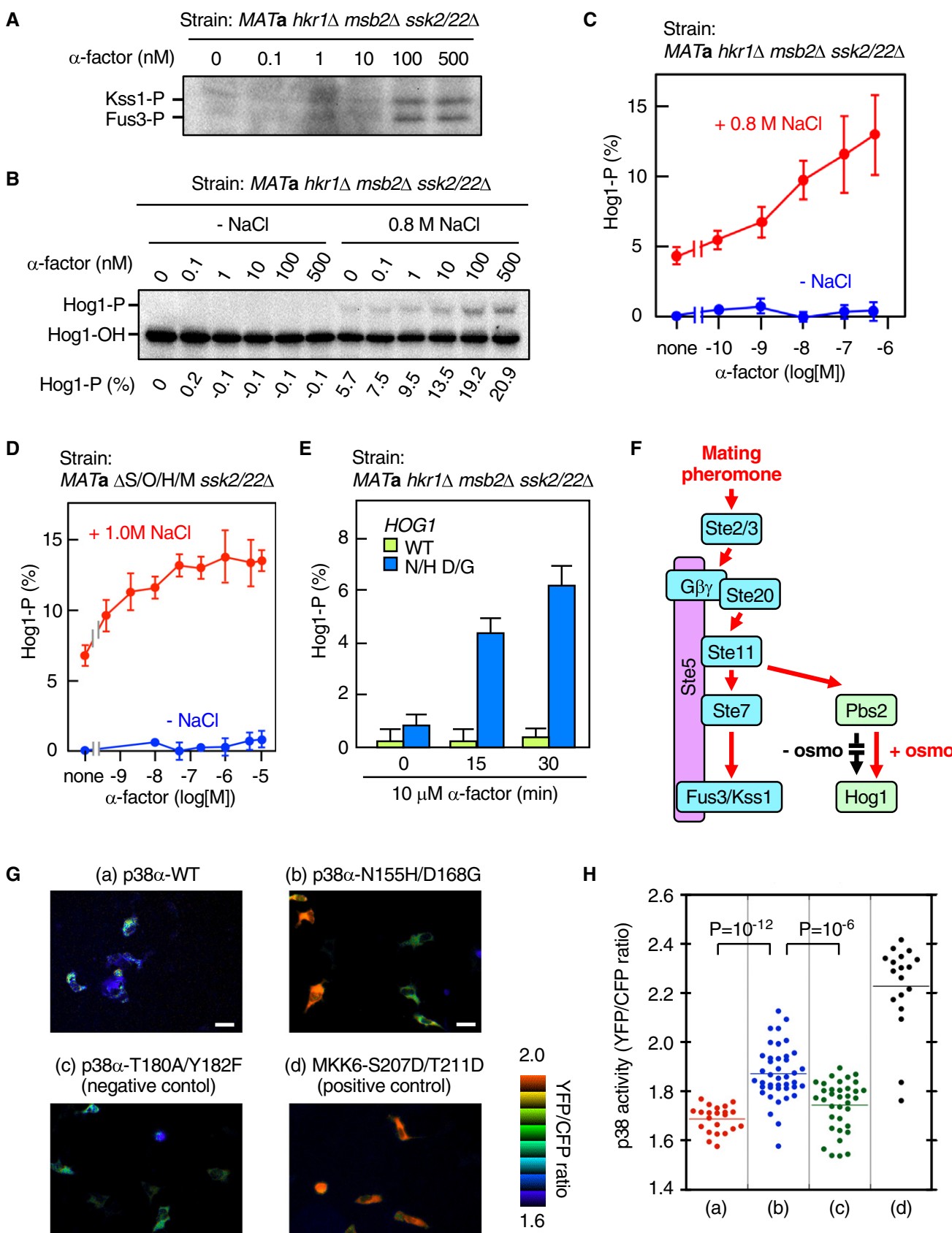

**Figure 8.**

**Figure 8.  The yeast pheromone-to-Hog1 crosstalk and the basal activity of the mammalian p38 MAPK might be suppressed by the same mechanism.**

A   KT299 (*MAT***a** *hkr1Δ msb2Δ ssk2/22Δ*) was exposed to the indicated concentrations of α-factor for 15 min in the absence of osmostress. Phosphorylation of Fus3 and Kss1 was detected by immunoblotting.

B   KT299 (*MAT***a** *ssk2/22Δ hkr1Δ msb2Δ*) was exposed to the indicated concentrations of α-factor for 15 min in the presence or absence of 0.8 M NaCl, and Hog1 phosphorylation was determined using the Phos-tag band-shift assay.

C   Average values of four independent experiments from (B) were plotted.

D   HM06-1 (*MAT***a** *ΔS/O/H/M ssk2/22Δ*) was exposed to the indicated concentrations (log scale) of α-factor for 15 min in the presence or absence of 1.0 M NaCl, and Hog1 phosphorylation was determined using the Phos-tag band-shift assay. Average values of three or more independent experiments were plotted.

E   KT306 (*MAT***a** *hkr1Δ msb2Δ ssk2/22Δ hog1Δ*) was transformed with pRS416-Hog1 (WT) or pRS416-Hog1-N149H D162G (N/H D/G) and was exposed to 10 μM α-factor for the indicated time in the absence of osmostress, and Hog1 phosphorylation was determined using the Phos-tag band-shift assay. Average of three independent experiments is plotted.

F   A schematic model showing that the lack of osmotic enhancement of the Pbs2-Hog1 reaction prevents the pheromone-to-Hog1 crosstalk.

G   Typical FRET (YFP/CFP ratio) images showing p38 activation. HeLa cells carrying the p38 reporter PerKy-p38 (Tomida *et al*, 2015) were stably transfected with an expression vector for the indicated p38α mutant proteins. FRET analysis was performed as described in Materials and Methods.

H   Distribution of p38 activity in individual cells from sets (a)–(d) in (G).

Data information: (B) Representative results from three independent experiments. (C–E) Error bars are SEM: (C) *n* = 4; (D), *n* = 3 or more; and (E) *n* = 3. (G) Scale bars: 20 μm. (H) Statistics, Student's two-tailed *t*-test.
Source data are available online for this figure.

between the modes was gradual, and in between the modes, the distinction was not necessarily clear. In WT cells, the three modes occurred at 0.2–0.8 M NaCl, 1.0–1.2 M NaCl, and > 1.4 M NaCl (Fig 9D). The WT profile is consistent with the numerous earlier reports (Van Wuytswinkel *et al*, 2000; Muzzey *et al*, 2009; Miermont *et al*, 2013; Babazadeh *et al*, 2014; English *et al*, 2015).

**Delayed Hog1 activation at severely high osmolarity is caused by a positive feedback mechanism**

The rapid-and-transient kinetics of the WT cells at the moderately high osmolarity is known to be caused by a negative feedback by accumulation of internal glycerol (Schaber *et al*, 2012), which is induced by activated Hog1 (Brewster *et al*, 1993; Albertyn *et al*, 1994). Increased internal glycerol concentrations reduce the difference between the internal and external osmotic pressures, thus effectively terminating the Hog1 activation process. In fact, in a mutant cell that has only kinase-dead Hog1-K52S/K53N or a *gpd1Δ gpd2Δ stl1Δ* mutant cell that can neither synthesize nor uptake glycerol (Ansell *et al*, 1997; Ferreira *et al*, 2005), the rapid activation occurs normally, but the subsequent decline is absent (for kinase-dead mutant) or much slower (for glycerol mutant; Fig 9E, left panel).

Interestingly, we found that no delayed activation occurred in Hog1-K52S/K53N or a *gpd1Δ gpd2Δ stl1Δ* mutant cells, suggesting

that the delayed activation was also caused by Hog1-induced glycerol accumulation (Fig 9E, right panel). In other words, Hog1-dependent reduction of osmostress causes both the positive and negative feedback regulations (Fig 9F). In the Discussion section, we propose a simple model that explains the appearance of the three kinetic modes by a common feedback mechanism.

# Discussion

**Possible mechanisms of the osmotic enhancement of the Pbs2-Hog1 reaction**

It is currently unknown how osmostress enhances the Pbs2-Hog1 reaction. However, our data significantly narrow the range of possibilities. Among the possible mechanisms, likely ones are as follows: (i) inhibition of the protein tyrosine phosphatases Ptp2/Ptp3 that dephosphorylate Hog1, (ii) enhancement of the Pbs2 activity, (iii) enhancement of the Pbs2-Hog1 interaction, and (iv) enhancement of the ability of Hog1 to be phosphorylated by Pbs2. We have excluded the inhibition of the phosphatases Ptp2/Ptp3 as the possible mechanism (see Fig EV2). The possibility that osmostress enhances the activity of Pbs2 has not been definitively excluded, but is unlikely as the osmotic and mutational enhancements of Hog1 phosphorylation could occur in the

**Figure 9.  Time courses and feedback regulation of the Hog1 phosphorylation.**

A   An example of the time-course experiments for the osmostress-induced Hog1 phosphorylation. The yeast strain KY603-3 (*ΔS/O/H/M ssk2/22Δ STE11-Q301P*) was stimulated with 1.0 M NaCl for the indicated times, and the percentage of phosphorylated Hog1 (Hog1-P) was determined using a Phos-tag band-shift assay.

B–D   Compilations of the time courses of osmostress-induced Hog1 activation in (B) KY603-3 (*ΔS/O/H/M ssk2/22Δ STE11-Q301P*), (C) TM257 (*ssk2/22Δ*), and (D) TM142 (wild-type). For clarity, time-course curves are shown in two panels for lower and higher ranges of NaCl concentrations. The color chart below (D) indicates the concentrations of NaCl used.

E   Effects of Hog1 kinase activity and osmostress-induced glycerol accumulation on the time courses of Hog1 activation. Cells were stimulated with 0.4 M NaCl (left panel) or 1.6 M NaCl (right panel) for the indicated times, and the percentage of phosphorylated Hog1 was determined using Phos-tag band-shift assay. Strains used were as follows: TM142 (WT), KT254 (*gpd1Δ gpd2Δ stl1Δ*), and TM232 (*hog1Δ*) carrying pRS416-HOG1-K52S K53N. K/S K/N, K52S K53N); *gpd1/2, gpd1Δ gpd2Δ*.

F   A model of the negative and positive feedback regulations in the HOG pathway. The upstream osmosensors are Sln1 and the complexes composed of Sho1, Opy2, Hkr1, and Msb2. The molecular identity of the downstream osmosensor, which enhances the signaling between Pbs2 and Hog1, is currently unknown.

Data information: (B–D) For each data point, *n* = 1 or more. (E) Error bars are SEM (*n* = 3 or more).
Source data are available online for this figure.

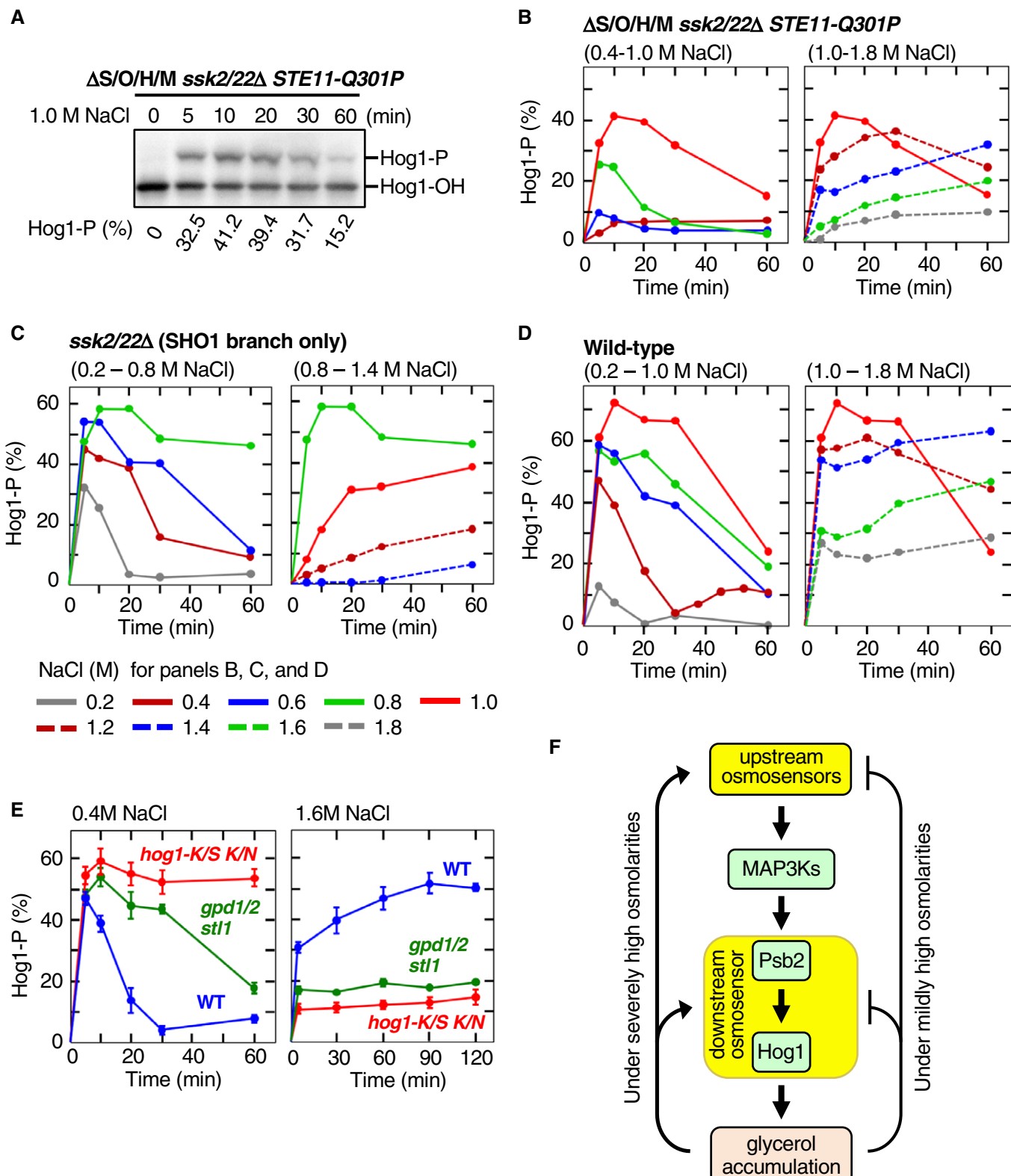

Figure 9.

absence of Pbs2 (Maayan *et al*, 2012) (also see Fig 7D). For the same reason, it is unlikely that osmostress enhances the Pbs2-Hog1 interaction. Our preliminary attempts to detect an osmotic enhancement of the Pbs2-Hog1 co-precipitation were unsuccessful.

Thus, we believe it most likely that osmostress enhances the ability of Hog1 to be phosphorylated by Pbs2. Osmostress might act directly on Hog1, or might do so through another, unidentified, protein.

To gain a further insight into how osmostress enhances the ability of Hog1 to be phosphorylated, we considered the structural aspects of the N149H/D162G and ΔL16 mutations. D162 is in the conserved DFG motif, which forms a flexible hinge at the amino-terminus of the activation loop (Fig EV4A). In the Hog1-homologous mammalian p38α (Wang *et al*, 1997), the residues corresponding to Hog1-N149 and D162 (p38-N155 and D168) are only 5.7 Å apart and are likely to affect each other's conformation (Fig EV4B). The DFG motif of the protein kinases including p38 is highly mobile and can assume either DFG-out or DFG-in conformations (Pargellis *et al*, 2002; Shan *et al*, 2009). Thus, a possible effect of the N149H/D162G mutation, and, by inference, of osmostress, is to induce a conformational change of the activation loop of Hog1 so that Hog1 becomes a better substrate of Pbs2 (Fig EV5A).

In contrast, Hog1-ΔL16 deletion mutation prevents osmotic enhancement of Hog1 phosphorylation. Although the Hog1-ΔL16 mutant was not activated by the basal activities (see Fig 7E, red), the Hog1-N149H/D162G-ΔL16 mutant was strongly phosphorylated (Fig 7E, orange). Thus, N149H/D162G mutation enhances the Pbs2-Hog1 reaction independently of the L16 region, suggesting the following order of signal transmission in the Hog1 molecule:

osmostress → L16 → N149/D162 → phosphorylation sites.

In the mammalian p38α, the L16 domain wraps the neck region and the N-lobe on the other side of the catalytic cleft (Fig EV4C) (Wang *et al*, 1997). As the flexible DGF motif is located in the neck region on the cleft side, a change at the neck region might affect the conformations of the DGF motif and the connected activation loop. In Hog1, osmostress might induce a conformational change in L16, which then secondarily induces a conformational change in the activation loop.

L16 is known to have diverse regulatory roles that are specific to each subfamily of MAPK. For example, binding of TAB1 to the L16 domain of mammalian p38 induces p38 auto-phosphorylation (Ge *et al*, 2002). Also, p38 mutants in L16 have been isolated that induces its auto-phosphorylation (Diskin *et al*, 2007). Clearly, a better understanding requires direct structural analyses of Hog1 itself.

### Crosstalk among the signaling pathways that share the MAP3K Ste11

Ste11 participates in three different MAPK cascades, namely the HOG pathway (Ste11-Pbs2-Hog1), the pheromone pathway (Ste11-Ste7-Fus3/Kss1), and the filamentous growth (FG) pathway (Ste11-Ste7-Kss1). In this report, we showed that the osmotic enhancement of the Hog1 phosphorylation (or rather, a lack of enhancement in the absence of osmostress) prevents the pheromone-to-Hog1 crosstalk. Although we have not examined in this work, it likely also prevents the FG-to-Hog1 crosstalk.

Conversely, osmostress fails to induce phosphorylation of the pheromone-specific MAPK Fus3 (Hao *et al*, 2008). Gβγ-bound Ste5, which is generated only when the mating pheromones are present, not only tethers the three kinases (Ste11, Ste7, and Fus3) together, but converts Fus3 to a better substrate of Ste7 (Good *et al*, 2009; Zalatan *et al*, 2012). In the absence of Gβγ-bound Ste5, activated Ste7 cannot phosphorylate Fus3 efficiently. Thus, there is a clear parallel between the Fus3 activation by pheromones and Hog1 activation by osmostress, namely that these kinases are converted by

their respective stimulus to a better substrate of their upstream MAP2K (Fig EV5B). This is also consistent with the finding by Patterson *et al* (2010) that there is no cross-inhibition between the pheromone MAPK pathway and the Hog1 MAPK pathway.

Unlike Fus3, the FG-specific MAPK Kss1 does not require Ste5 binding for its phosphorylation by Ste7 (Flatauer *et al*, 2005). Thus, an important barrier is absent for the crosstalk activation of Kss1 by osmostress. In fact, osmostress activates Kss1 by a rapid-and-transient kinetics similar to that of Hog1 activation (Hao *et al*, 2008; Shock *et al*, 2009; Nishimura *et al*, 2016). Failure of osmostress to induce the FG pathway reporter gene *TEC1-lacZ* (Shock *et al*, 2009) is likely due to the brief period of Kss1 activation, which perhaps is insufficient to induce the reporter gene.

### Physiological roles of the osmotic enhancement of the Hog1 phosphorylation

In addition to the prevention of the pheromone-to-Hog1 crosstalk, the osmotic enhancement of Hog1 phosphorylation has at least two more important functions. First, it prevents Hog1 activation by basal activities of the MAP3Ks. This role is of particular importance for the SLN1 pathway, as the phospho-relay mechanism in the SLN1 pathway is intrinsically prone to basal activation of the MAP3Ks Ssk2/Ssk22. The activator of Ssk2/22, namely Ssk1, is phosphorylated at Asp-554 when there is no osmostress (Posas *et al*, 1996). Osmostress induces dephosphorylation of Ssk1-P to Ssk1-OH, which then binds and activates the MAP3Ks Ssk2/22 (Posas & Saito, 1998; Horie *et al*, 2008). Since phospho-aspartate undergoes spontaneous hydrolysis (Janiak-Spens *et al*, 1999), there always exists some dephosphorylated Ssk1-OH in unstimulated cells, hence high basal activities of Ssk2/Ssk22. High basal activity is important for rapid response of the SLN1 branch to osmostress (Macia *et al*, 2009).

Second, it expands the range of osmolarity to which the HOG pathway can respond. As shown in Fig 1I, Hog1 activation at severely high osmolarities is almost exclusively mediated by the SLN1 branch. Perhaps because S514 phosphorylation is weak at extremely high osmolarities (see Fig 5G), enhancement-defective Hog1ΔL16 cannot be phosphorylated at extremely high osmolarities (Fig 4). In other words, the osmotic enhancement of the Pbs2-Hog1 reaction likely extends the higher limit of osmolarity to which the HOG pathway can respond.

### A model for the delayed Hog1 phosphorylation at severely high osmolarities

Delayed phosphorylation of Hog1 at severely high osmolarity has been observed previously (Babazadeh *et al*, 2013; Miermont *et al*, 2013). The slow phosphorylation kinetics at severely high osmolarity has been interpreted as caused by cell volume decrease and molecular crowding, but the details remained vaguely defined.

Here, we propose a new model based on a few observations and a simple assumption.

1. We observed that Hog1-dependent accumulation of glycerol is necessary for the delayed Hog1 activation (see Fig 9E).
2. We also observed that delayed Hog1 activation occurs at different osmolarities among different strains (see Figs 9B–D and EV5C–E).

3   As previously reported, Hog1 dephosphorylation is rapid, and the extent of Hog1 phosphorylation closely follows the changes in external osmolarities (Mettetal *et al*, 2008; Zi *et al*, 2010).

4   Finally, we assume that the levels of Hog1 phosphorylation at 5 min (as reported in Fig 1I) are good approximations of the signaling efficiencies that are not yet modulated by feedback.

From these facts and assumption, we can deduce the following scenario. When extracellular NaCl concentration ($[NaCl]_{ext}$) is below the optimal concentration (i.e., the NaCl concentration that activates Hog1 most strongly at 5 min), the gradually increasing intracellular glycerol concentration ($[gly]_{int}$) lowers the osmotic pressure and rapidly weakens the response (Fig EV5C–E, blue). When $[NaCl]_{ext}$ is slightly higher than the optimal, lowering of the osmotic pressure by glycerol accumulation does not substantially change the Hog1 response, which would explain the appearance of plateau (Fig EV5C–E, red). Finally, when $[NaCl]_{ext}$ is significantly higher than the optimal, lowering of the osmotic pressure by glycerol accumulation would increase the Hog1 response, causing a delayed activation (Fig EV5C–E, green). Thus, the time courses of Hog1 activation at any initial external osmolarity can be explained by the common feedback loop shown in Fig 9F. It also explains why delayed activation occurs at different osmolarities for strains with different complement of osmosensors, which might be difficult to explain by the molecular crowding alone.

### Possible evolutionary origin of the three-kinase MAPK cascades

The MAPK cascades consist of three kinases. Evolutionarily, complex systems must have developed from less complex systems, by addition of a new element, by duplication of a part of the system, etc. Here, we briefly outline a possible evolutionary path that might have generated the current three-kinase MAPK pathway. It has been shown that a mutant of Hog1 can auto-activate (auto-phosphorylate) upon osmostress in the absence of the upstream MAP2K (Maayan *et al*, 2012). In this case, Hog1 itself might be considered as an osmosensor, and such a "one-kinase, one-osmosensor" system might have been the primordial form of the HOG pathway (Fig EV5F). Addition of a constitutively active MAP2K, such as Pbs2-DD, generates a "two-kinase, one-osmosensor" system that can activate Hog1 more efficiently. Our results (see Fig 2C, lanes 7–9) demonstrate that such a two-kinase system can be constructed. Finally, addition of a MAP3K and upstream osmosensors generates the modern "three-kinase, two-osmosensor" system that is even more efficient and has a better osmostress-specificity than the simpler systems. Replacement of the upstream osmosensors by other sensors/receptors can easily generate the MAPK cascades of different specificities.

## Materials and Methods

### Media

YPD medium consists of 10 mg ml$^{-1}$ yeast extract (Nacalai Tesque), 20 mg ml$^{-1}$ tryptone (Nacalai Tesque), and 20 mg ml$^{-1}$ glucose. CAD medium consists of 6.7 mg ml$^{-1}$ yeast nitrogen base (Sigma-Aldrich), 20 mg ml$^{-1}$ glucose, 5 mg ml$^{-1}$ Bacto™ Casamino Acids (BD), and appropriate supplements (20 µg ml$^{-1}$ uracil and 40 µg ml$^{-1}$ tryptophan) as needed.

### Reagents

5-bromo-4-chloro-3-indolyl-β-D-galactoside (X-gal) was from Takara Bio. The stock solution contained 20 mg ml$^{-1}$ X-gal in *N, N*-dimethylformamide. Phos-tag acrylamide AAL-107 was purchased from Wako Pure Chemical. The stock solution contained 5.0 mM Phos-tag acrylamide AAL-107 in 3% (v/v) methanol. Bis(2-hydroxyethyl)iminotris(hydroxymethyl)methane (Bis-Tris) was from Wako Pure Chemical. Other chemicals were purchased from Sigma-Aldrich, Wako Pure Chemical, Nacalai Tesque, and BD.

### Buffers

Buffer A contained 50 mM Tris–HCl (pH 7.5), 15 mM EDTA, 15 mM EGTA, 2 mM dithiothreitol (DTT), 1 mM phenylmethylsulfonyl fluoride (PMSF), 1 mM benzamidine, 5 µg ml$^{-1}$ leupeptin, 150 mM NaCl, and 0.2% (v/v) Triton X-100. Buffer P contained 50 mM Tris–HCl (pH 7.5), 2 mM DTT, 1 mM PMSF, 1 mM benzamidine, and 5 µg ml$^{-1}$ leupeptin. Tris-buffered saline (TBS) contained 25 mM Tris–HCl (pH 7.4), 137 mM NaCl, and 2.68 mM KCl. Buffer Z (for β-galactosidase assay) contained 60 mM Na$_2$HPO$_4$, 40 mM NaH$_2$PO$_4$, 10 mM KCl, and 1 mM MgSO$_4$, adjusted to pH 7.0. X-gal/Buffer Z contained 0.33 mg ml$^{-1}$ X-gal and 38 mM 2-mercaptoethanol in Buffer Z. SDS loading buffer (1×) contained 50 mM Tris–HCl (pH 6.8), 2% SDS, 0.1 mg ml$^{-1}$ Bromophenol Blue, 10% (v/v) glycerol, and 700 mM 2-mercaptoethanol.

### Yeast strains

All yeast mutants used in this work are derivatives of the S288C strain (Table 1).

### Plasmid constructs

Deletion and missense mutants were constructed using PCR-based oligonucleotide mutagenesis (Ho *et al*, 1989) and were confirmed by nucleotide sequence determination. The following plasmids were used.

#### Vector plasmids
pRS414 (= *TRP1*, *CEN6*), pRS416 (= *URA3*, *CEN6*), and YCplac22I' (= *TRP1*, *CEN4*) are yeast single-copy plasmids with the indicated nutritional selective markers (Sikorski & Hieter, 1989; Tatebayashi *et al*, 2003).

#### Ste11 expression plasmids
pRS416-Ste11 (= $P_{STE11}$-*STE11*, *URA3*, *CEN6*) and its mutant derivatives are *STE11* genomic DNA clones that express Ste11 under the control of the *STE11* promoter.

#### Pbs2 expression plasmids
YCplac22I'-Pbs2 (= $P_{PBS2}$-*PBS2*, *TRP1*, *CEN4*) and its mutant derivatives are *PBS2* genomic DNA clones that express Pbs2 under the

control of the *PBS2* promoter. YCplac22I'-Pbs2-HA and its mutant derivatives encode C-terminally HA-tagged version of YCplac22I'-Pbs2. pRS414-FLAG-Pbs2 (= $P_{PBS2}$-FLAG-PBS2, TRP1, CEN6) is a *PBS2* genomic DNA clone that expresses the N-terminally FLAG-tagged Pbs2 under the control of the *PBS2* promoter.

### Hog1 expression plasmids

pRS416-Hog1 (= $P_{HOG1}$-HOG1, URA3, CEN6) and its mutant derivatives are *HOG1* genomic DNA clones that express Hog1 under the control of the *HOG1* promoter. pRS416-FLAG-Hog1 and its mutant derivatives are N-terminally FLAG-tagged version of pRS416-Hog1.

### Hog1 reporter plasmids

pRS414-8xCRE-lacZ (= 8xCRE-lacZ, TRP1, CEN6) is a Hog1-specific gene expression reporter, and pRS414-8xCRE-CYC^m-lacZ (= 8xCRE-CYC^m*lacZ*, TRP1, CEN6) is its attenuated version. Both plasmids have been described in detail (Tatebayashi *et al*, 2006).

### MKK6 expression plasmid

pcDNA3-Myc-MKK6DD encodes the N-terminally Myc-tagged constitutively active MKK6-S207D/T211D (Takekawa *et al*, 1998).

### p38 expression plasmid

pcDNA3-FLAG-p38α encodes the N-terminally FLAG-tagged murine p38 alpha. pcDNA3-FLAG-p38α-T180A/Y182F (Addgene plasmid #20352) was a gift from Dr. Roger Davis (Enslen *et al*, 1998). pcDNA3-FLAG-p38α-WT and pcDNA3-FLAG-p38α-N155H/D168G were constructed by modifying pcDNA3-FLAG-p38α-T180A/Y182F by PCR-based oligonucleotide mutagenesis.

### p38 activity reporter plasmid

The FRET-based p38 MAPK activity reporter PerKy-p38 has been described (Tomida *et al*, 2015).

### Hog1-specific reporter assay

Reporter assays using the Hog1-specific reporter plasmid pRS414-8xCRE-lacZ have been described (Tatebayashi *et al*, 2006). All reporter assays were carried out in triplicate (or more) using independent cultures. Activity of lacZ (β-galactosidase) was expressed in Miller units (Miller, 1972) and is presented as an average value and standard error of the mean (SEM).

### Yeast culture and osmostress treatment

Yeast cells not carrying plasmids were grown in YPD at 30°C. Cells carrying plasmids were grown in CAD with appropriate supplements at 30°C. For osmostress treatment, fresh overnight cultures (in YPD or CAD) were diluted in YPD to $OD_{600}$ = 0.25 and cultivated with vigorous aeration until the $OD_{600}$ reached 0.6–0.8. 6 ml (for Hog1) or 15 ml (for Pbs2-HA) of culture was centrifuged at 2,000 *g* for 3 min. The supernatant was removed by aspiration, and the cell pellets were resuspended in the original volume of warm (30°C) YPD containing an appropriate concentration of NaCl or sorbitol. Cells were further incubated at 30°C until 3 min before the desired time, when cells were subjected to centrifugation at 2,000 *g* for 3 min at 30°C. The supernatant was removed by aspiration, and the cell pellets were resuspended in 1 ml of Buffer P and were

centrifuged briefly, and the supernatant was removed by aspiration. Cell pellets were immediately frozen in liquid nitrogen and stored at −80°C until use.

### Cell extract preparation

Frozen cell pellets were resuspended in 400 μl of Buffer A. Cell suspensions were ground vigorously using glass beads at 4°C, and then centrifuged at 9,200 *g* for 10 min at 4°C to sediment cell debris. The supernatants (cell extracts) were either immediately used for immunoprecipitation or SDS–PAGE analyses, or frozen in liquid nitrogen and stored at −80°C until use.

### Immunoprecipitation

For immunoprecipitation of FLAG-Hog1 and its mutant derivatives, the cell extracts were incubated with 30 μl of anti-FLAG M2 Affinity gel (Sigma-Aldrich) for 2 h at 4°C. The affinity gels were washed three times with buffer A, resuspended in 1× SDS loading buffer, and boiled for 5 min. The gels were sedimented by brief centrifugation, and the supernatants were loaded for SDS–PAGE.

For immunoprecipitation of Pbs2-HA and its mutant derivatives, the cell extracts were incubated with 50 μl of Protein G Sepharose beads (GE Healthcare) for 2 h at 4°C, following the addition of 1 μg of anti-HA antibody 3F10 (Roche) and incubation for 1 h at 4°C. Beads were washed 3 times with buffer A, resuspended in 1× SDS loading buffer, and boiled for 5 min. The beads were sedimented by brief centrifugation, and the supernatants were loaded for SDS–PAGE or Phos-tag SDS–PAGE.

### Immunoblotting

After electrophoresis (SDS–PAGE, or Phos-tag SDS–PAGE), proteins were transferred onto Amersham™ Protran™ 0.45 μm nitrocellulose membranes (GE Healthcare) or onto Amersham™ Hybond ™ P 0.45 μm PVDF membranes (GE Healthcare). Phosphorylated Hog1 was detected by immunoblotting using the anti-phospho-p38 MAPK (T180/Y182) antibody #9211 (Cell Signaling Technology). Hog1 was detected using the anti-Hog1 antibody yC-20 (Santa Cruz Biotechnology). FLAG-Hog1 was detected by anti-FLAG M2 antibody (Sigma-Aldrich). Pbs2-HA was detected by anti-HA antibody F-7 (Santa Cruz Biotechnology). Phosphorylation of Pbs2 Thr-518 was detected by polyclonal anti-Pbs2 phospho-T518 antibody custom produced by SCRUM Inc. Phosphorylated Kss1 and Fus3 were detected using the anti-phospho-p44/42 MAPK (Erk1/2; T202/Y204) Rabbit monoclonal antibody #4370 (Cell Signaling Technology).

### Anti-Pbs2 phospho-T518 antibody

Anti-Pbs2 phospho-T518 antibody was custom produced by SCRUM Inc. As the antigen, the phosphopeptide LAK(pT)NIGCQS, which corresponds to the Pbs2 residues 515–524, was synthesized and coupled to KLH at the peptide C terminus. Two rabbits were immunized five times with the antigen mixed with the Freund's complete adjuvant. The serum from one of the rabbits was subjected to affinity purification using a column conjugated with the phosphopeptide LAK(pT)NIGCQS. The bound antibodies were eluted from the column and passed through another column conjugated with

**Table 1.  Yeast strains used in this study.**

| Strain | Genotype | Source |
|---|---|---|
| FP4 | MAT**a** ura3 leu2 trp1 his3 hog1::TRP1 | F. Posas |
| FP54 | MAT**a** ura3 leu2 trp1 his3 ste11::HIS3 | Posas et al (1998) |
| HM06-1 | MAT**a** ura3 leu2 trp1 his3 ssk2::LEU2 ssk22::LEU2 sho1::hisG opy2::natMX4 hkr1::hphMX4 msb2::kanMX6 | This study |
| KT003 | MAT**a** ura3 leu2 trp1 his3 pbs2::LEU2 | This study |
| KT005 | MAT**a** ura3 leu2 trp1 his3 ste11::HIS3 pbs2::LEU2 | Tatebayashi et al (2003) |
| KT043 | MAT**a** ura3 leu2 trp1 his3 ssk2::hisG ssk22::hisG ste11::kanMX6 pbs2::HIS3 | This study |
| KT207 | MATα ura3 leu2 trp1 his3 ssk2::LEU2 ssk22::LEU2 pbs2::hphMX4 | This study |
| KT209 | MATα ura3 leu2 trp1 his3 ssk2::LEU2 ssk22::LEU2 sho1::hisG opy2::natMX4 hkr1::hphMX4 msb2::kanMX6 STE11-Q301P pbs2::URA3 | This study |
| KT219 | MATα ura3 leu2 trp1 his3 ssk2::LEU2 ssk22::LEU2 sho1::hisG opy2::natMX4 hkr1::hphMX4 msb2::kanMX6 ste11::HIS3MX6 | This study |
| KT234 | MATα ura3 leu2 trp1 his3 ssk2::LEU2 ssk22::LEU2 sho1::hisG opy2::natMX4 hkr1::hphMX4 msb2::kanMX6 ste11::HIS3MX6 pbs2::URA3 | This study |
| KT235 | MATα ura3 leu2 trp1 his3 ssk2::LEU2 ssk22::LEU2 sho1::hisG opy2::natMX4 hkr1::hphMX4 msb2::kanMX6 STE11-Q301P hog1::HIS3MX6 | This study |
| KT248 | MATα ura3 leu2 trp1 his3 ssk2::LEU2 ssk22::LEU2 sho1::hisG opy2::natMX4 hkr1::hphMX4 msb2::kanMX6 pbs2::LEU2 hog1::HIS3MX6 | This study |
| KT250 | MATα ura3 leu2 trp1 his3 ssk2::LEU2 ssk22::LEU2 sho1::hisG opy2::natMX4 hkr1::hphMX4 msb2::kanMX6 hog1::HIS3MX6 | This study |
| KT254 | MAT**a** ura3 leu2 trp1 his3 gpd1::natMX4 gpd2::hphMX4 stl1::kanMX6 | This study |
| KT259 | MAT**a** ura3 leu2 trp1 his3 ste11::HIS3 hog1::natMX4 | This study |
| KT260 | MAT**a** ura3 leu2 trp1 his3 ssk2::hisG ssk22::hisG ste11::kanMX6 hog1::natMX4 | This study |
| KT290 | MAT**a** ura3 leu2 trp1 his3 ssk2::hisG ssk22::hisG ste11::kanMX6 pbs2::HIS3 hog1::natMX4 | This study |
| KT291 | MAT**a** ura3 leu2 trp1 his3 ste11::HIS3 pbs2::hphMX4 hog1::natMX4 | This study |
| KT292 | MATα ura3 leu2 trp1 his3 ssk1::LEU2 ste11::hphMX4 hog1::TRP1 | This study |
| KT293 | MATα ura3 leu2 trp1 his3 ssk2::LEU2 ssk22::LEU2 sho1::natMX4 hog1::hphMX4 | This study |
| KT294 | MATα ura3 leu2 trp1 his3 ssk2::LEU2 ssk22::LEU2 opy2::kanMX6 hog1::hphMX4 | This study |
| KT295 | MATα ura3 leu2 trp1 his3 ssk2::LEU2 ssk22::LEU2 hkr1::natMX4 msb2::kanMX6 hog1::hphMX4 | This study |
| KT299 | MAT**a** ura3 leu2 trp1 his3 ssk2::hisG ssk22::hisG hkr1::hphMX4 msb2::kanMX6 | This study |
| KT303 | MATα ura3 leu2 trp1 his3 ssk2::LEU2 ssk22::LEU2 sho1::hisG opy2::natMX4 hkr1::hphMX4 msb2::kanMX6 STE11-Q301P ptp2::HIS3MX6 | This study |
| KT305 | MATα ura3 leu2 trp1 his3 ssk2::LEU2 ssk22::LEU2 sho1::hisG opy2::natMX4 hkr1::hphMX4 msb2::kanMX6 STE11-Q301P ptp2::URA3 ptp3::HIS3MX6 | This study |
| KT306 | MAT**a** ura3 leu2 trp1 his3 ssk2::hisG ssk22::hisG hkr1::hphMX4 msb2::kanMX6 hog1:: natMX4 | This study |
| KT307 | MATα ura3 leu2 trp1 his3 ssk2::LEU2 ssk22::LEU2 sho1::hisG opy2::natMX4 hkr1::hphMX4 msb2::kanMX6 STE11-Q301P ptp3::HIS3MX6 | This study |
| KY523 | MAT**a** ura3 leu2 trp1 his3 ssk2::hisG ssk22::hisG hog1::LEU2 | Yamamoto et al (2010) |
| KY594-1 | MATα ura3 leu2 trp1 his3 ssk2::LEU2 ssk22::LEU2 sho1::hisG opy2::natMX4 hkr1::hphMX4 msb2::kanMX6 | Tatebayashi et al (2015) |
| KY603-3 | MATα ura3 leu2 trp1 his3 ssk2::LEU2 ssk22::LEU2 sho1::hisG opy2::natMX4 hkr1::hphMX4 msb2::kanMX6 STE11-Q301P | This study |
| TM142 | MATα ura3 leu2 trp1 his3 | T. Maeda |
| TM232 | MATα ura3 leu2 his3 hog1::LEU2 | T. Maeda |
| TM257 | MATα ura3 leu2 trp1 his3 ssk2::LEU2 ssk22::LEU2 | Tatebayashi et al (2006) |
| TM280 | MAT**a** ura3 leu2 trp1 ssk2::LEU2 ssk22::LEU2 pbs2::URA3 | Maeda et al (1995) |
| YM105 | MAT**a** ura3 leu2 trp1 his3 pbs2::kanMX6 hog1::LEU2 | Murakami et al (2008) |

unphosphorylated peptide LAKTNIGCQS to eliminate the antibodies that bound to the unphosphorylated peptide.

### Phos-tag SDS–PAGE

Phos-tag SDS–PAGE was conducted essentially as described previously (Kinoshita & Kinoshita-Kikuta, 2011; English *et al*, 2015). The separating gel contained 80 mg ml$^{-1}$ acrylamide/bisacrylamide (37.5:1), 350 mM Bis-Tris–HCl (pH 6.8), 40 μM (for Hog1) or 20 μM (for Pbs2) Phos-tag acrylamide AAL-107, and 80 μM ZnCl$_2$ (for Hog1) or 40 μM ZnCl$_2$ (for Pbs2). The stacking gel contained 40 mg ml$^{-1}$ acrylamide/bis acrylamide (37.5:1) and 350 mM Bis-Tris–HCl (pH 6.8). The running buffer consisted of 50 mM Tris, 50 mM MOPS, 1 mg ml$^{-1}$ SDS, and 5.0 mM sodium bisulfite.

### Phospho-Hog1 band-shift assay

Frozen cell pellets were resuspended in 250 μl of 1× SDS loading buffer. The cell suspensions were vigorously ground using glass beads at 4°C and centrifuged at 9,200 *g* for 10 min at 4°C. The supernatants (cell lysates) were either immediately used, or frozen in liquid nitrogen and stored at −80°C. Aliquots (12 μl) of the cell lysates were mixed with 10 μl of 1× SDS loading buffer, boiled for 5 min, and separated by Bis-tris SDS–PAGE containing 40 μM Phos-tag (Phos-tag SDS–PAGE). After electrophoresis, the bound phosphopeptides were released from Phos-tag acrylamide by soaking the gel first in a solution consisting of 25 mM Tris, 192 mM glycine, 20% (v/v) methanol, and 1.0 mM EDTA for 15 min twice, and then in a solution consisting of 25 mM Tris, 192 mM glycine, and 20% (v/v) methanol for 20 min. The proteins in the gel were transferred onto Amersham™ Hybond ™ P 0.45 μm PVDF membranes (GE Healthcare). Hog1 was detected by immunoblotting using the anti-Hog1 antibody yC20 (Santa Cruz Biotechnology). The slow- and fast-migrating Hog1 bands represent phosphorylated and unphosphorylated forms, respectively. Enhanced chemiluminescence images were digitally captured using the ChemiDoc XRS Plus (Bio-Rad) equipped with a charge-coupled-device camera. Quantitation of band intensity was carried out using the Image Lab program (version 4.1, Bio-Rad).

### Phospho-Pbs2 band-shift assay

Phospho-Pbs2 band-shift assay was conducted essentially as phospho-Hog1 band-shift assay except that Pbs2-HA was first concentrated by immunoprecipitation from cell extracts before subjected to Phos-tag SDS–PAGE.

### Mass spectrometry

The cell extracts for the mass spectrometric analyses of Pbs2 phosphorylation were prepared as follows. The yeast strains TM280 (*ssk2/22Δ pbs2Δ*) and KT005 (*ssk2/22Δ pbs2Δ*) were transformed with pRS414-FLAG-Pbs2, which encoded N-terminally FLAG-tagged Pbs2. Cells grown in CAD medium overnight were inoculated into YPD medium at OD$_{600}$ = 0.25. Cells were then cultured for 4 h with vigorous shaking. NaCl was added to final conc. of 0.6 M (or not added for a negative control) to the cell cultures, and immediately, cells were subjected to centrifugation at 1,930 *g* for 3 min at 30°C.

Cell pellets were collected and resuspended in ice-cold Buffer P (these processes took 7 min after the NaCl addition). Cells were again centrifuged at 20,630 *g* for 5 s at 4°C, and cell pellets were frozen in liquid nitrogen. Cells were resuspended in Buffer A, mixed with glass beads (approximately a half volume of the cell suspension), and ground by three rounds of 5 min vortexing at 4°C with a 2-min cooling period in between. The lysates were cleared of cell debris by centrifugation at 9,200 *g* for 10 min at 4°C. To immunoprecipitate the FLAG-tagged Pbs2, the cell extracts were incubated with 50 μl of anti-FLAG M2 Affinity gel (Sigma-Aldrich) for 2 h at 4°C. The affinity gels were washed three times with buffer A, and two additional times with TBS. The bound proteins were eluted from the gel with 100 μl TBS containing 0.2 mg ml$^{-1}$ FLAG peptide (Sigma-Aldrich, # F3290) for 1 h on ice. The eluates were separated from the gel by centrifugation at 8,200 *g* for 1 min at 4°C twice. A portion of the eluate was subjected to SDS–PAGE, and co-precipitated proteins were visualized using the SilverQuest™ Silver staining kit (Invitrogen).

The remainder of the eluate was trypsin-digested, desalted using ZipTip C18 (Millipore), and centrifuged in a vacuum concentrator. Shotgun proteomic analyses were performed by a linear ion trap-orbitrap mass spectrometer (LTQ-Orbitrap Velos, Thermo Fisher Scientific) coupled with a nanoflow LC system (Dina-2A, KYA Technologies) (Hirano *et al*, 2013). Peptides were injected into a 75-μm reversed-phase C$_{18}$ column at a flow rate of 10 μl min$^{-1}$ and eluted with a linear gradient of solvent A (2% acetonitrile and 0.1% formic acid in H$_2$O) to solvent B (40% acetonitrile and 0.1% formic acid in H$_2$O) at 300 nl min$^{-1}$. Peptides were sequentially sprayed from a nanoelectrospray ion source (KYA Technologies) and analyzed by collision-induced dissociation (CID). The analyses were operated in data-dependent mode, switching automatically between MS and MS/MS acquisition. For CID analyses, full-scan MS spectra (from *m/z* 380 to 2,000) were acquired in the orbitrap with a resolution of 100,000 at *m/z* 400 after ion count accumulation to the target value of 1,000,000. The 20 most intense ions at a threshold above 2,000 were fragmented in the linear ion trap with a normalized collision energy of 35% for an activation time of 10 ms. The orbitrap analyzer was operated with the "lock mass" option to perform shotgun detection with high accuracy. Protein identification was conducted by searching MS and MS/MS data using Mascot ver. 2.5.1 (Matrix Science). Methionine oxidation, protein N-terminal acetylation, pyro-glutamination for N-terminal glutamine, and phosphorylation of serine, threonine, and tyrosine were set as variable modifications. A maximum of two missed cleavages was allowed in our database search, while the mass tolerance was set to three parts per million (ppm) for peptide masses and 0.8 Da for MS/MS peaks. In the process of peptide identification, we applied a filter to satisfy a false discovery rate lower than 1%.

### Isolation of *HOG1* mutants that mimic the osmotic enhancement

Initially, a *HOG1* mutant library was generated using an error-prone PCR method. pRS416-Hog1 (*URA3* marker) was doubly digested with the restriction enzymes EcoRI and SnaBI to obtain a gapped plasmid DNA that lacked most of the *HOG1* coding DNA sequence (CDS). A DNA segment from pRS416-Hog1 containing the entire *HOG1* CDS and the flanking regions was amplified by PCR in the presence of 0.2 mM MnCl$_2$. The gapped plasmid DNA and the

mutagenized PCR products were co-transformed into KT235 (*sho1Δ opy2Δ hkr1Δ msb2Δ ssk2Δ ssk22Δ hog1Δ STE11-Q301P*) that harbored a plasmid encoding an attenuated *8xCRE-lacZ* reporter gene (pRS414-8xCRE-CYC^m-lacZ; *TRP1* marker). Cells containing both a gap-repaired plasmid (*URA3*$^+$) and the reporter plasmid (*TRP1*$^+$) were selected on CAD plates (w/o uracil and tryptophan). Formed colonies were replica-plated onto nitrocellulose membrane disks. To permeabilize the cells, the disks were frozen in liquid N$_2$ for 30 s and thawed at room temperature. The disks were then soaked in X-gal/Buffer Z and incubated at 30°C for 1–12 h. *HOG1* mutants that induced *8xCRE-lacZ* expression and rendered the colonies blue were selected as candidates that encode constitutively enhanced Hog1. These candidates were then individually transformed into KT235 (*STE11-Q301P*) and KT250 (*STE11-WT*), and expression of the Hog1-reporter gene *8xCRE-lacZ* was assayed.

### Cell culture and transfection

HEK293A cells were maintained in Dulbecco's modified Eagle's medium supplemented with 10% fetal bovine serum, L-glutamine, penicillin, and streptomycin. For imaging analyses, cells were plated on a 35-mm-diameter glass bottom dish (MatTek, P35GC-0-14-C) at 30% confluency. Transfection was carried out on the same day using Effectene (Qiagen), in accordance with the manufacturer's protocol.

All imaging analyses were carried out at 37°C under 5% CO$_2$.

### FRET imaging analysis

Activity of the p38 MAPK was measured using the FRET-based p38 activity probe PerKy-p38, essentially as described previously (Tomida *et al*, 2015). Fluorescence images of the cells were captured using the Nikon TE-2000E inverted microscope equipped with the CFI PlanApoVC 20X (NA 0.75) objective, a CoolSnapHQ CCD camera (Roper), a heat-CO$_2$ chamber, a mercury lamp (Nikon), and a computer-controlled emission filter changer (Ludl Electronic Products). A FRET filter set (Semrock) was used to excite the fluorescent proteins at 440 nm, and to acquire fluorescence images at 480 nm (for CFP) and at 535 nm (for YFP). Images were acquired using MetaMorph software (Molecular Devices). ImageJ software was used to measure mean fluorescence intensities of individual cells. To select the region of interest, entire cell region was manually located based on CFP fluorescence image. Pixel-by-pixel ratio images were generated using the MetaMorph software and were displayed with an intensity-modulated (IMDisplay) mode based on the YFP fluorescence intensity.

Expanded View for this article is available online.

### Acknowledgements

We thank Drs. Yoshikazu Nakamura and Sumiko Watanabe for generously allowing us to use their space and equipment, Dr. Pauline O'Grady for critical editing of the manuscript, and Ms Makiko Umeda for her administrative assistance. This work was supported in part by JSPS Grants-in-Aid for Scientific Research (KAKENHI) Grant Numbers 16H04761 (to K.T.), 25440042 (to K.Y.), 16H06578 (to M.O.), 17H06017 and 19K06548 (to T.Tomida), 17K15083 (to Y.T.), and 24247034 (to H.S.); by grants from Sumitomo Foundation (number 150075), Salt Science Research Foundation (number 1732), and the Japan Foundation for Applied Enzymology to K.T.; and by a grant from the Mochida Memorial Foundation for Medical and Pharmaceutical Research to T.Tomida. A.N. was supported by the Graduate Program for Leaders in Life Innovation from MEXT.

### Author contributions

KT and HS conceived the project; KT, KY, TTo, AN, TTa, and HS designed experiments; KT, KY, TTo, AN, and TTa performed the experiments; KT, MO, and HK-H designed and performed mass Spectrometric analyses; and SA-A and YT contributed critical ideas. All authors were involved in analyzing data and preparing the manuscript.

### Conflict of interest

The authors declare that they have no conflict of interest.

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
