## [Review Process File · The EMBO Journal]

Osmostress enhances activating phosphorylation of Hog1 MAP kinase by mono-phosphorylated Pbs2 MAP2K

Kazuo Tatebayashi, Katsuyoshi Yamamoto, Taichiro Tomida, Akiko Nishimura, Tomomi Takayama, Masaaki Oyama, Hiroko Kozuka-Hata, Satomi Adachi-Akahane, Yuji Tokunaga, and Haruo Saito

Review timeline:

Submission date:	11th Sep 2019
Editorial Decision:	18th Oct 2019
Revision received:	29th Nov 2019
Editorial Decision:	19th Dec 2019
Revision received:	22nd Dec 2019
Accepted:	8th Jan 2020

Editor: Ieva Gailite

Transaction Report:

1st Editorial Decision

18th Oct 2019

Thank you for submitting your manuscript for consideration by the EMBO Journal. We have now received two referee reports on your manuscript, which are included below for your information.

As you will see from the comments, both reviewers appreciate the work and the topic. However, they also raise a number of concerns that need to be addressed before they can support publication here. From my side, I judge the referee comments to be generally reasonable, therefore, based on the overall interest expressed in the reports, I would like to invite you to submit a revised version of your manuscript in which you address the comments of both referees. I should add that it is The EMBO Journal policy to allow only a single major round of revision and that it is therefore important to resolve the main concerns at this stage.

REFeree REPORTS:

Referee #1:

Review of "Osmotic priming of the MAPK Hog1 and an AND-gate mechanism prevent non-osmotic activation of Hog1" by Tatebayashi et al.

In this manuscript the authors make the remarkable finding that the MAP3Ks (Ste11 and Ssk2), which on the HOG pathway converge on the MAP2K Pbs2, can phosphorylate Pbs2 in one of the two sites only (Ste11 only in T518 while Ssk2/22 in either S514 or T518), but not both sites when cells are subjected to "suboptimal" osmolarity stress conditions. In these conditions, mono phosphorylated Pbs2 cannot phosphorylate the MAPK Hog1. However, they discover that during osmotic stress, by an unknown mechanism they term "priming", there is a change that allows Hog1 to be activated by this monophosphorylated Pbs2. They further isolate Hog1 mutants that cannot be

primed (Hog1- Δ (320-350)) or are "constitutively primed" (Hog1-N149H/D162G). Despite these findings, the paper does not explain the nature of the priming process, even though there is a section in which they speculate as to what could be going on with Hog1, but it is pure speculation. Thus, the manuscript blends careful molecular biology details with less satisfying "black box" aspects, such as the concept of "priming".

Major comments:

In the manuscript, they made several interesting claims, some of which need extra experiments or clarifications:

1- In general, the experiments shown would greatly benefit from repeating them (or selected ones) as time courses (as opposed to the final time point assays presented across all this paper), since it has been shown that at higher osmolarities the overall response is delayed and peak phosphorylation takes place at different times. The complete absence of time courses in this manuscript obscures the interpretation of the results. For example, a time course might answer my concern number 2, below.

2- The authors show that Hog1 can be activated in the absence of the known osmosensors, in response to high osmolarity shock, directly by Ste11. Curiously, there is a biphasic dose response to osmolarity (with a peak around 1M NaCl, depending on the deletion strain used). What is the nature of this biphasic behavior?

3- The dose-response obtained with only the Sho1 branch is also biphasic (green curve in 1H). Do the authors have an explanation for a lack of phosphorylation at very high osmolarity? Could this be related to the biphasic response in the absence of osmosensors when only Ste11 is present? That is, is it possible that Ste11 itself is osmosensitive and cannot "work" at those high osmolarities? (assuming the biphasic response is not explained by delayed dynamics, as I suggest above).

4- The L16 mutant, which cannot be "primed" is especially deficient at being phosphorylated at high osmolarity, just as it happens in the Δ skk2/22 strain with WT Hog1. Thus, it seems that the Δ -L16 mutant cannot be phosphorylated specifically by the Sln1 branch. This observation led the author to test if Ssk2 and Ste11 act differently on Pbs2, and indeed, even though Ssk2 can phosphorylate both S514 and T518, Ste11 can do it only on T518.

Missing from this analysis, surprisingly, is a dose response of WT cells stimulated at very high osmolarities, the region where in the deletion strains phosphorylation drops down (above 1M NaCl). Why is that? That would be informative, since it might show a collaboration (synergy) between the branches.

5- Related to the above concern, the authors do not seem to show (as the abstract conveys), that Pbs2 ever exists in WT cells in monophosphorylated form. This is a critical piece of information, since they claim that in the absence of high osmolarity, the "unprimed" Hog1 cannot be activated by monophosphorylated Pbs2, but di-phosphorylated Pbs2 can. Since the Sln1 branch seems to be able to generate di-phosphorylated Pbs2, it is important to determine if mono-phosphorylated Pbs2 exists in normal cells.

6- The authors propose that osmostress acts on Hog1 directly, since even a constitutively active Pbs2-DD mutant needs osmostress to be able to phosphorylate Hog1 (Fig 2D-E). However, they cannot rule out with this experiment that osmostress acts directly on Pbs2 (instead of on top of Hog1), producing a conformational change on Pbs2 (or causing the interaction by a third protein).

7- The authors say that mono-phosphorylated Pbs2 can only phosphorylate Hog1 when Hog1 is primed by osmolarity. They based that on the inability of Pbs2 to act on the Hog1- Δ -L16, which they called "unprimable". There is no mechanistic explanation of the priming effect and why in this mutant this process would fail to happen. I suggest them to rephrase in general the use of the word priming. Here for example, they have a nice mutant that cannot be phosphorylated by Pbs2 mono-phosphorylated. It would be interesting to understand why. But saying that that is because Hog1 cannot be "primed" is not illuminating, it creates the sense that one knows what is the problem with that Hog1, when one really doesn't know.

8- In the cross-talk experiment, it would be important to test a strain with the triple deletion of Sho1 and ssk2/22, besides the quadruple one they do test: hkr1 msb2 ssk2/22. This is because the pheromone pathway could act via Sho1 instead of via Ste11, which is the whole point of the experiment. For example, one of the genes induced by the mating pathway, Fus1, is known to interact with Sho1.

About the cross-talk, if the authors are correct, then pheromone stimulation should be able to activate the "constitutively primed" mutant of Hog1 in the absence of high osmolarity in the medium. The authors should do this easy experiment, since it will be very informative.

9- In the discussion, there is little comparison with other work. Is there none? I recall a paper a few years ago, by the Colman-Lerner group, that showed that pheromone can activate the Hog1 pathway in high osmolarity, but if I'm not mistaken, that was dependent on an active Sln1 branch. Is there any relationship with the results in this paper?

10- Also in the discussion, the authors propose that one function of osmotic priming is to prevent unwanted activation by other signals (similar to what they show for the cross-talk with pheromone). This seems to contradict a lot of previous literature. First, there are a number of other stimuli that activate Hog1 kinase, besides high osmolarity and arsenite, which the authors seem to neglect. Second, these stimuli, usually rely on one or the other branches, and they do not need the presence of high osmolarity. Just two examples: acetic acid uses the SLN1 branch, and zymolyase uses the Sho1 branch. Thus, if the authors are correct, how is it that a stress that uses only one branch can activate Hog1 in the absence of osmotic stress "priming".

10- The section of the results called "A two-step activation mechanism of the Hog1 MAPK" should be moved to the discussion, since it does not really contain results, it is their proposed model of Hog1 priming.

1- The discussion in general is disappointing, given the remarkable findings of the paper, one would expect a stronger discussion, commenting on how this finding fits in in the vast literature of MAPK cascades.

Referee #2:

In this study the authors investigate new mechanistic questions underlying activation of the yeast MAPK Hog1 by hyperosmotic stimuli. They find that signal transmission through this kinase cascade is considerably more complex than previously appreciated. In particular, osmotic stress provides a "second input", at a downstream position in the pathway, separate from the osmosensor-induced activation of the upstream MAP3Ks. A diverse series of elegant experiments uncover numerous new insights, including that the second input has a unique dose-response profile, that it acts downstream of Pbs2 phosphorylation, and that it depends on a distinct sequence region (L16) in Hog1. Moreover, they demonstrate that both residues in the activation loop of Pbs2 are phosphorylated, that either mono-phosphorylated form can at least partly signal to Hog1, and that the upstream MAP3Ks differ in their ability to phosphorylate these sites. Finally, the authors identify a mutant form of Hog1 (N149H/D162G) that behaves as if it is constitutively "primed", leading to elevated Hog1 phosphorylation under basal conditions and susceptibility to crosstalk. The overall findings lead to a model in which the second input primes Hog1 to be a better substrate for its activator, thus constituting an "AND-gate" that enhances signaling under inefficient conditions and helps maintain pathway specificity.

Overall, this is an insightful, high-quality study. The experiments are extraordinarily thorough, the results are clear and convincing, and the overall interpretations are compelling. The findings are surprising and thought provoking, and the presentation clearly highlights both the advances in mechanistic understanding as well as the potential physiological benefits for the new features uncovered. Ultimately, I find no serious faults, as the work has comprehensively and convincingly addressed the most relevant, key issues. I think the manuscript could be published largely as-is. I do have a variety of specific comments, listed below, that I suggest would be useful to address and/or incorporate into this already excellent manuscript. None require further experimentation, and can be addressed by additions or clarifications in the text.

Specific points.

1. The evidence for a "second input" is clear. The authors suggest that it acts at the level of Hog1, and they eventually conclude that osmostress "primes" Hog1 in a way that makes it "a better substrate of Pbs2". Strictly speaking, this is not really demonstrated here but rather is an interpretation of the results. I think it is a reasonable interpretation, and perhaps the simplest, but it is not the only possible interpretation. An obvious alternative is that the "primed" condition makes Hog1 a poorer substrate for inactivating phosphatases; this could apply to both the effect of osmostress and the effect of the N149H/D162G mutant. Another would be that osmostress stimulates Pbs2 activity in a manner distinct from changes in its phosphorylation state. Another would be that osmostress enhances the Pbs2 → Hog1 phosphorylation reaction by increasing their local concentration. Therefore, I think it would be valuable (in the Discussion) for the authors to discuss such alternatives directly, describe whether the data favors some over others, and perhaps comment on future approaches to resolve any remaining ambiguities (e.g., *in vitro* kinase assays in the absence of counteracting phosphatases, etc.). Doing so would make clear to readers the specific logic that the authors used to settle upon their favored interpretation.

2. I think the authors should explicitly mention and cite prior precedent that signaling output downstream of activated Ste11 is increased by stimulus in both the HOG and pheromone pathways, as reported previously (Lamson et al 2006 [PMID: 16546088]; Tatebayashi et al 2006 [PMID: 16778768]). Relevant places to cite these would be in the Results, when describing the findings that osmostress can stimulate signal output "at a point downstream of MAP3Ks" (pg 8, middle) or where commenting on the role of the AND-gate in preventing crosstalk (pg 17, line 7), or at related parts of the Discussion (pg 20-21). It seems fair and appropriate to do so, and this would not in any way detract from the numerous additional insights in this paper.

3. Page 21, bottom: "The conversion from the unprimed state to the primed state makes Hog1 a better substrate of Pbs2." There is a highly analogous situation in the pheromone pathway, in that efficient activation of the MAPK Fus3 requires a second input that allows the MAPK to be a better substrate of its MAP2K, Ste7 (Good et al 2009 [PMID: 19303851] and Zalatan et al 2012 [PMID: 22878499]). It is certainly worth citing these studies in the Discussion and noting both the functional parallels as well as mechanistic similarities vs. differences.

4. In Figure 5G, the negative values plotted for Hog1 Δ L16 cannot be correct; the lowest possible value is zero. Something must be wrong with the calculation or normalization procedure, or perhaps with the method of background subtraction. Presumably the Hog1-P signal was undetectable, and hence it should be zero, not negative.

5. Page 14, middle: "... indicating that mono-phosphorylated Pbs2 could phosphorylate Hog1 (Figure 5B, lanes 3-6)." Here it would be useful to add a direct comment that the LEVEL of Hog1-P is not equal to WT for either Pbs2 mutant (i.e., WT max = 50-60%; T518A max = 30%; S514A max = 9%).

Other minor points.

6. For all of the dose-response plots throughout the paper, the zero molar NaCl data point is excluded from the graphs, despite being included in all the western blots. Why is this? I could not think of a good reason to exclude it, and I found no explanation. In some cases (e.g., Fig 3G, 3I) it partly obscures the fact that the low-osmolarity values do indeed reflect an induced response rather than a basal level.

7. Page 14, top: "Comparisons of Figure 4G with 3G, and Figure 4H with 3I, suggested that only the conditions that strongly generated phosphorylated S514 could activate Hog1 without osmotic priming." This description is a bit confusing. It would help to provide a clear and explicit description of what key feature readers should examine when making these "comparisons". Based on the parenthetical comment in the subsequent sentence, I suspect the authors wish to draw attention to the fact that S514 phosphorylation occurs only in *ste11 Δ* strains and at low-to-medium osmolarity, and that these are also conditions in which the Δ L16 mutation does not compromise Hog1 phosphorylation. Is that correct? If so, perhaps state this directly.

8. Page 15, line 7: "...Ste11-Q301P (which produces only mono-phosphorylated Pbs2)". I could not find where this claim was demonstrated. Perhaps this is being inferred based on the finding in Fig 4 that Ste11-WT in *ssk2/22Δ* mutants does not phosphorylate S514 of Pbs2. If so, it would be useful to clarify that this is the basis of the claim.

9. The Fig 6A legend needs some description of what is represented by each set of 3 lanes. I suspect these are independent samples done in triplicate.

10. Page 15, 3rd line from bottom: typo - "therefor"

1st Revision - authors' response

29th Nov 2019

Point-by-point responses to reviewers' comments

Note: In the revised manuscript, changes and additions made specifically in response to the referees' comments are highlighted by blue. Other, mostly minor, changes are not highlighted.

In the revised manuscript, many figure numbers are changed, and new figures are added. The figure numbers of newly added experimental data are listed below.

- Fig 4A-B Phosphorylation of hog1-ΔL16 in WT strain**
- Fig 5J Analysis of mono- and di-phosphorylated Pbs2 in WT strain**
- Fig 7D Activity of auto-activating Hog1 mutant**
- Fig 8D Crosstalk experiment using a *sho1Δ* strain (ΔS/O/H/M *ssk2/22Δ*)**
- Fig 8E Crosstalk experiment using the Hog1-N149H D162G mutant**
- Fig 9A-D Time-course experiments**
- Fig 9E Demonstration of feedback regulation**
- Fig EV2 Test for the involvement of Ptp2/Ptp3 in osmotic enhancement**

Referee #1:

In this manuscript the authors make the remarkable finding that the MAP3Ks (Ste11 and Ssk2), which on the HOG pathway converge on the MAP2K Pbs2, can phosphorylate Pbs2 in one of the two sites only (Ste11 only in T518 while Ssk2/22 in either S514 or T518), but not both sites when cells are subjected to "suboptimal" osmolarity stress conditions. In these conditions, mono phosphorylated Pbs2 cannot phosphorylate the MAPK Hog1. However, they discover that during osmostress, by an unknown mechanism they term "priming", there is a change that allows Hog1 to be activated by this monophosphorylated Pbs2. They further isolate Hog1 mutants that cannot be primed (Hog1-Δ(320-350)) or are "constitutively primed" (Hog1-N149H/D162G). Despite these findings, the paper does not explain the nature of the priming process, even though there is a section in which they speculate as to what could be going on with Hog1, but it is pure speculation. Thus, the manuscript blends careful molecular biology details with less satisfying "black

box" aspects, such as the concept of "priming".

Response: We thank the referee for the positive and very constructive comments on our manuscript. We have addressed all the points as described below.

Major comments:

In the manuscript, they made several interesting claims, some of which need extra experiments or clarifications:

1- In general, the experiments shown would greatly benefit from repeating them (or selected ones) as time courses (as opposed to the final time point assays presented across all this paper), since it has been shown that at higher osmolarities the overall response is delayed and peak phosphorylation takes place at different times. The complete absence of time courses in this manuscript obscures the interpretation of the results. For example, a time course might answer my concern number 2, below.

Response: We added complete sets of time-course experiments for three strains ($\Delta S/O/H/M$ *ssk2/22 Δ hog1 Δ STE11-Q301P*, the SHO1 branch only strain *ssk2/22 Δ* and wild-type) in Fig 9A-9D. As the referee correctly predicted, Hog1 phosphorylation at severely high osmolarities was delayed in all the three strains. A newly added section entitled "Time-courses of the Hog1 phosphorylation at various osmolarities" (pages 19-20) describes the results.

2- The authors show that Hog1 can be activated in the absence of the known osmosensors, in response high osmolarity shock, directly by Ste11. Curiously, there is a biphasic dose response to osmolarity (with a peak around 1M NaCl, depending on the deletion strain used). What is the nature of this biphasic behavior?

Response: We added a set of experiments that showed that the delayed Hog1 activation at severely high osmolarities is caused by a positive feedback regulation induced by Hog1 activation and glycerol accumulation (Fig 9E-9F). We also showed that the rapid-and-transient Hog1 activation at mildly high osmolarities is controlled by a negative feedback regulation, as previously well known. A newly added section entitled "Delayed Hog1 activation at severely high osmolarities is caused by a positive feedback mechanism" (pages 20-21) describes the results.

3- The dose-response obtained with only the Sho1 branch is also biphasic (green curve in 1H). Do the authors have an explanation for a lack of phosphorylation at very high osmolarity? Could this be related to the biphasic response in the absence of osmosensors when only Ste11 is present? That is, is it possible that Ste11 itself is osmosensitive and cannot "work" at those high osmolarities? (assuming the

biphasic response is not explained by delayed dynamics, as I suggest above).

Response: From the experimental data we obtained in response to the referee's first and second comments, we reached a new explanation of the delayed Hog1 activation at severely high osmolarities (not just for the newly found downstream osmosensing, but for the HOG pathway activation in general). We added a new section in the Discussion entitled "A model for the delayed Hog1 phosphorylation at severely high osmolarities" (pages 25-26), in which we proposed a new model that would explain how both the negative feedback at mildly high osmolarities and the positive feedback at severely high osmolarities are governed by the same mechanism involving glycerol accumulation.

4- The L16 mutant, which cannot be "primed" is especially deficient at being phosphorylated at high osmolarity, just as it happens in the delta ssk2/22 strain with WT Hog1. Thus, it seems that the delta-L16 mutant cannot be phosphorylated specifically by the Sln1 branch. This observation led the author to test if Ssk2 and Ste11 act differently on Pbs2, and indeed, even though Ssk2 can phosphorylate both S514 and T518, Ste11 can do it only on T518.

Missing from this analysis, surprisingly, is a dose response of WT cells stimulated at very high osmolarities, the region where in the deletion strains phosphorylation drops down (above 1M NaCl). Why is that? That would be informative, since it might show a collaboration (synergy) between the branches.

Response: We added a complete dose response data for the Hog1- Δ L16 phosphorylation in WT cells (Fig 4A-B). The WT response was very similar to that of the SLN1-only strain (*ste11 Δ*). In particular, at the severely high osmolarities, WT responses were very low, indicating that there was no significant synergy between the SLN1 and SHO1 branches.

5- Related to the above concern, the authors do not seem to show (as the abstract conveys), that Pbs2 ever exists in WT cells in monophosphorylated form. This is a critical piece of information, since they claim that in the absence of high osmolarity, the "unprimed" Hog1 cannot be activated by monophosphorylated Pbs2, but di-phosphorylated Pbs2 can. Since the Sln1 branch seems to be able to generate di-phosphorylated Pbs2, it is important to determine if mono-phosphorylated Pbs2 exists in normal cells.

Response: We analyzed the phosphorylation status of Pbs2 in the WT cells, using the combination of Phos-tag band-shift and anti-T518P immunoblotting (Fig 5J). The results for WT cells are very similar to the SLN1-only strain (*ste11 Δ*) shown in Fig 5I. In the absence of osmostress, the basal activities of Ssk2/Ssk22 likely generate mono-phosphorylated Pbs2, but the amount is too low for unambiguous detection by this approach. In contrast, at severely high osmolarities (e.g., 1.6 M NaCl), there clearly is mono-phosphorylated Pbs2 (phosphorylated only at T518).

Because the presence of mono-phosphorylated Pbs2 in unstressed cells is a deduction rather than an observed fact, we changed a sentence in the abstract to the following, more accurate, statement, "Here we report that the MAP3K Ste11 phosphorylates only one activating phosphorylation site (Thr-518) in Pbs2, whereas the MAP3Ks Ssk2/Ssk22 can phosphorylate both Ser-514 and Thr-518 under optimal osmostress conditions."

6- The authors propose that osmostress acts on Hog1 directly, since even a constitutively active Pbs2-DD mutant needs osmostress to be able to phosphorylate Hog1 (Fig 2D-E).

However, they cannot rule out with this experiment that osmostress acts directly on Pbs2 (instead or on top of Hog1), producing a conformational change on Pbs2 (or causing the interaction by a third protein).

Response: This is essentially the same comment as the comment 1 of ref #2. We fully agree with the referees that our results did not provide any evidence that osmostress directly acted on Hog1. As both the referees had the same concern, we gave a very close attention to this point in the revised manuscript. Specifically, we considered the following four possibilities for the osmotic enhancement of Hog1 phosphorylation.

- 1) Inhibition of the protein tyrosine phosphatases Ptp2/Ptp3
- 2) Enhancement of the Pbs2 activity
- 3) Enhancement of the Pbs2-Hog1 interaction
- 4) Enhancement of the ability of Hog1 to be phosphorylated by Pbs2

To test the model 1, we added the new data (Fig EV2), in which we showed that the osmotic enhancement of Hog1 phosphorylation occurs even in a *ptp2Δ ptp3Δ* double mutant. Thus, inhibition of Ptp2/Ptp3 is excluded from the possibilities.

The models 2 and 3 are made less likely, though perhaps not completely eliminated, by the experiment shown in Fig 7D. We showed that the Hog1-F318S/H344L mutant can weakly auto-activate in the absence of Pbs2, which is consistent with the previous report that Hog1-F318S can auto-phosphorylate (Maayan *et al* (2012)). The Hog1-F318S mutant is believed to have a slight catalytic activity in the absence of the phosphorylation at the TEY motif, and Hog1-F318S serves both as a kinase and its own substrate. Thus, phosphorylation of Hog1-F318S by Hog1-F318S occurs, though very weakly. In this context, we found that a combination of the auto-activating mutation F318S/H344L and the activation enhancing mutation N149H/D162G generated a Hog1 mutant that auto-phosphorylates extremely strongly in the absence of Pbs2. This result suggests that the enhancing effect of N149H/D162G is not specific to Pbs2, rendering the models 2 and 3 unlikely.

Thus, by elimination, we consider the model 4 most likely. However, even if the model 4 is correct, it is possible that the effect is mediated by a third protein that interacts with (or modifies) Hog1. We added these considerations

in the section in the Discussion entitled "Possible mechanisms of the osmotic enhancement of the Pbs2-Hog1 reaction" (the first paragraph; page 22).

7- The authors say that mono phosphorylated Pbs2 can only phosphorylate Hog1 when Hog1 is primed by osmolarity. They based that on the inability of Pbs2 to act on the Hog1-delta-L16, which they called "unprimable". There is no mechanistic explanation of the priming effect and why in this mutant this process would fail to happen. I suggest them to rephrase in general the use of the word priming. Here for example, they have a nice mutant that cannot be phosphorylated by Pbs2 mono phosphorylated. It would be interesting to understand why. But saying that that is because Hog1 cannot be "primed" is not illuminating, it creates the sense that one knows what is the problem with that Hog1, when one really doesn't know.

Response: We used the word "priming" only as a convenient shorthand for an unknown mechanism, but it seemed to have caused more confusion than clarification. We therefore replaced all occurrences of "priming" (and related phrases) by more neutral word/phrases such as "osmotic enhancement of the Pbs2-Hog1 reaction." Similarly, "constitutively primed Hog1" was replaced by "constitutively-enhanced Hog1." The title of the paper was also changed accordingly.

8- In the cross-talk experiment, it would be important to test a strain with the triple deletion of Sho1 and ssk2/22, besides the quadruple one they do test: hkr1 msb2 ssk2/22. This is because the pheromone pathway could act via Sho1 instead of via Ste11, which is the whole point of the experiment. For example, one of the genes induced by the mating pathway, Fus1, is known to interact with Sho1.

Response: We agree that the involvement of Sho1, and perhaps of Opy2, was a real possibility. We thus repeated the crosstalk experiment using a $\Delta S/O/H/M$ ssk2/22 Δ strain (in which both Sho1 and Opy2 are deleted). As shown in Fig 8D, crosstalk does take place in this strain as well as in the hkr1 msb2 ssk2/22 strain. Thus, we could exclude the involvement of Sho1 and Opy2 in the crosstalk activation of Hog1.

About the cross-talk, if the authors are correct, then pheromone stimulation should be able to activate the "constitutively primed" mutant of Hog1 in the absence of high osmolarity in the medium. The authors should do this easy experiment, since it will be very informative.

Response: As predicted by the referee, we found that the "constitutively-enhanced" Hog1 N149H D162G mutant was phosphorylated upon pheromone stimulation in the absence of osmostress. This is shown in Fig 8E.

9- In the discussion, there is little comparison with other work. Is there none? I recall a paper a few years ago, by the Colman-Lerner group, that showed that pheromone can activate the Hog1 pathway in high osmolarity, but if I'm not

mistaken, that was dependent on an active Sln1 branch. Is there any relationship with the results in this paper?

Response: In Discussion (pages 23-24), we added a section entitled "Crosstalk among the signaling pathways that share the MAPK Ste11," in which we briefly discussed the relevant findings concerning the crosstalk among the three MAPK cascades that share Ste11.

We are not aware of any report that showed a condition that breaks down the barrier for the pheromone-to-Hog1 crosstalk, except by artificial tethering of components from two pathways ("rewiring"). As the referee commented, Baltanás *et al* (Sci Sig, 6-ra26 (2013)) observed a pheromone-induced Hog1 activation. Although their work is very interesting, it is clear that it does not have any direct relationship to the pheromone-to-Hog1 crosstalk discussed in our paper. Baltanás *et al* observed that, when yeast cells were pre-adapted to 1M sorbitol overnight, pheromone addition activated Hog1 after 50 min. With detailed analyses, they clearly demonstrated that the following sequence of the events occurred:

- 1) Adaptation to 1 M sorbitol increases the intracellular glycerol concentration.
- 2) Pheromone activates Fus3, which triggers the shmoo formation.
- 3) The shmoo formation weakens the cell wall, and activates the MAPK Slt2/Mpk1.
- 4) Activated Slt2/Mpk1 opens the Fsp1 glycerol channel.
- 5) Glycerol efflux and the presence of 1 M sorbitol in the culture media creates a hyperosmotic condition.
- 6) Hog1 is activated.

In fact, the authors clearly stated in their Abstract that "Activation of HOG by the PR (pheromone response pathway) was not due to loss of insulation, but rather a response to a reduction in internal osmolarity." Citing this paper only serves to confuse the readers.

10- Also in the discussion, the authors propose that one function of osmotic priming is to prevent unwanted activation by other signals (similar to what they show for the cross-talk with pheromone). This seems to contradict a lot of previous literature. First, there are a number of other stimuli that activate Hog1 kinase, besides high osmolarity and arsenite, which the authors seem to neglect. Second, these stimuli, usually rely on one or the other branches, and they do not need the presence of high osmolarity. Just two examples: acetic acid uses the SLN1 branch, and zymolyase uses the Sho1 branch. Thus, if the authors are correct, how is it that a stress that uses only one branch can activate Hog1 in the absence of osmotic "priming"

Response: We realize that our statement was too broad. What we really meant was that if a non-osmotic stimulus activated one of the upstream osmosensors weakly, it would not induce significant Hog1 activation. We rewrote the section in Discussion (pages 24-25) entitled "Physiological roles of the osmotic enhancement of the Hog1 phosphorylation" to be more precise.

The referee commented a number of reported non-osmotic stimuli that can activate Hog1. Arsenite activates Hog1 by inhibiting Ptp2/Ptp3, without activating osmosensors. As we explained in the reply to comment 6, the osmotic enhancement of Hog1 activation does not involve the inhibition of these phosphatases.

Incidentally, many other "non-osmotic" stimuli might directly or indirectly induce osmostress. For example, 100 mM NaAcetate likely activates Hog1 just as 100 mM NaCl activates Hog1. At low pH, inhibition of glycerol accumulation might further enhance Hog1 phosphorylation by preventing negative feedback (FEMS Yeast Res 6:1274(2006)). Zymolyase might modify the cell wall and create osmotic imbalance, but experiments using zymolyase are problematic, because zymolyase preparations are usually contaminated with proteases (to different degrees). Thus, in our own experiments, contrary to others' report, zymolyase activated Hog1 dependent on the SLN1 branch (JCB, 161:1035 (2003)).

11- The section of the results called "A two-step activation mechanism of the Hog1 MAPK" should be moved to the discussion, since it does not really contain results, it is their proposed model of Hog1 priming.

Response: We have moved this section to the Discussion as a part of the section entitled "Possible mechanisms of the osmotic enhancement of the Pbs2-Hog1 reaction" (the second and third paragraphs; page 22-23).

12- The discussion in general is disappointing, given the remarkable findings of the paper, one would expect a stronger discussion, commenting on how this finding fits in in the vast literature of MAPK cascades.

Response: We completely reorganized and rewrote the Discussion section. We also added a new section entitled "Possible evolutionary origin of the three-kinase MAPK cascades" (page 26).

Referee #2:

In this study the authors investigate new mechanistic questions underlying activation of the yeast MAPK Hog1 by hyperosmotic stimuli. They find that signal transmission through this kinase cascade is considerably more complex than previously appreciated. In particular, osmotic stress provides a "second input", at a downstream position in the pathway, separate from the osmosensor-induced activation of the upstream MAP3Ks. A diverse series of elegant experiments uncover numerous new insights, including that the second input has a unique dose-response profile, that it acts downstream of Pbs2 phosphorylation, and that it depends on a distinct sequence region (L16) in Hog1. Moreover, they demonstrate that both residues in the activation loop of Pbs2 are phosphorylated, that either mono-phosphorylated form can at least partly signal to Hog1, and that the

upstream MAP3Ks differ in their ability to phosphorylate these sites. Finally, the authors identify a mutant form of Hog1 (N149H/D162G) that behaves as if it is constitutively "primed", leading to elevated Hog1 phosphorylation under basal conditions and susceptibility to crosstalk. The overall findings lead to a model in which the second input primes Hog1 to be a better substrate for its activator, thus constituting an "AND-gate" that enhances signaling under inefficient conditions and helps maintain pathway specificity.

Overall, this is an insightful, high-quality study. The experiments are extraordinarily thorough, the results are clear and convincing, and the overall interpretations are compelling. The findings are surprising and thought provoking, and the presentation clearly highlights both the advances in mechanistic understanding as well as the potential physiological benefits for the new features uncovered. Ultimately, I find no serious faults, as the work has comprehensively and convincingly addressed the most relevant, key issues. I think the manuscript could be published largely as-is. I do have a variety of specific comments, listed below, that I suggest would be useful to address and/or incorporate into this already excellent manuscript. None require further experimentation, and can be addressed by additions or clarifications in the text.

Response: We thank the referee for the positive and very constructive comments on our manuscript. We have addressed all the points as described below.

Specific points.

1. The evidence for a "second input" is clear. The authors suggest that it acts at the level of Hog1, and they eventually conclude that osmostress "primes" Hog1 in a way that makes it "a better substrate of Pbs2". Strictly speaking, this is not really demonstrated here but rather is an interpretation of the results. I think it is a reasonable interpretation, and perhaps the simplest, but it is not the only possible interpretation. An obvious alternative is that the "primed" condition makes Hog1 a poorer substrate for inactivating phosphatases; this could apply to both the effect of osmostress and the effect of the N149H/D162G mutant. Another would be that osmostress stimulates Pbs2 activity in a manner distinct from changes in its phosphorylation state. Another would be that osmostress enhances the Pbs2 → Hog1 phosphorylation reaction by increasing their local concentration. Therefore, I think it would be valuable (in the Discussion) for the authors to discuss such alternatives directly, describe whether the data favors some over others, and perhaps comment on future approaches to resolve any remaining ambiguities (e.g., *in vitro* kinase assays in the absence of counteracting phosphatases, etc.). Doing so would make clear to readers the specific logic that the authors used to settle upon their favored interpretation.

Response: This is essentially the same comment as the comment 6 of ref #1. We fully agree with the referees that our results did not provide any evidence that osmostress directly acted on Hog1. As both the referees had the same

concern, we gave a very close attention to this point in the revised manuscript. Specifically, we considered the following four possibilities for the osmotic enhancement of Hog1 phosphorylation.

- 1) Inhibition of the protein tyrosine phosphatases Ptp2/Ptp3
- 2) Enhancement of the Pbs2 activity
- 3) Enhancement of the Pbs2-Hog1 interaction
- 4) Enhancement of the ability of Hog1 to be phosphorylated by Pbs2

To test the model 1, we added the new data (Fig EV2), in which we showed that the osmotic enhancement of Hog1 phosphorylation occurs even in a *ptp2Δ ptp3Δ* double mutant. Thus, inhibition of Ptp2/Ptp3 is excluded from the possibilities.

The models 2 and 3 are made less likely, though perhaps not completely eliminated, by the experiment shown in Fig 7D. We showed that the Hog1-F318S/H344L mutant can weakly auto-activate in the absence of Pbs2, which is consistent with the previous report that Hog1-F318S can auto-phosphorylate (Maayan *et al* (2012)). The Hog1-F318S mutant is believed to have a slight catalytic activity in the absence of the phosphorylation at the TEY motif, and Hog1-F318S serves both as a kinase and its own substrate. Thus, phosphorylation of Hog1-F318S by Hog1-F318S occurs, though very weakly. In this context, we found that a combination of the auto-activating mutation F318S/H344L and the activation enhancing mutation N149H/D162G generated a Hog1 mutant that auto-phosphorylates extremely strongly in the absence of Pbs2. This result suggests that the enhancing effect of N149H/D162G is not specific to Pbs2, rendering the models 2 and 3 unlikely.

Thus, by elimination, we consider the model 4 most likely. However, even if the model 4 is correct, it is possible that the effect is mediated by a third protein that interacts with (or modifies) Hog1. We added these considerations in the section in the Discussion entitled "Possible mechanisms of the osmotic enhancement of the Pbs2-Hog1 reaction." (page 22).

2. I think the authors should explicitly mention and cite prior precedent that signaling output downstream of activated Ste11 is increased by stimulus in both the HOG and pheromone pathways, as reported previously (Lamson et al 2006 [PMID: 16546088]; Tatebayashi et al 2006 [PMID: 16778768]). Relevant places to cite these would be in the Results, when describing the findings that osmostress can stimulate signal output "at a point downstream of MAP3Ks" (pg 8, middle) or where commenting on the role of the AND-gate in preventing crosstalk (pg 17, line 7), or at related parts of the Discussion (pg 20-21). It seems fair and appropriate to do so, and this would not in any way detract from the numerous additional insights in this paper.

Response: We added the following comment immediately after the description of Fig 1D (page 7).

"It has been previously observed that the endogenous-level expression of constitutively-active Ste11 mutant does not activate Hog1 unless osmostress is applied (Lamson *et al.*, 2006, Tatebayashi *et al.*, 2006).."

3. Page 21, bottom: *"The conversion from the unprimed state to the primed state makes Hog1 a better substrate of Pbs2."* There is a highly analogous situation in the pheromone pathway, in that efficient activation of the MAPK Fus3 requires a second input that allows the MAPK to be a better substrate of its MAP2K, Ste7 (Good *et al* 2009 [PMID: 19303851] and Zalatan *et al* 2012 [PMID: 22878499]). It is certainly worth citing these studies in the Discussion and noting both the functional parallels as well as mechanistic similarities vs. differences.

Response: In the section in Discussion entitled "Crosstalk among the signaling pathways that share the MAPK Ste11," we added the following comments (pages 23-24).

"Conversely, osmostress fails to induce phosphorylation of the pheromone-specific MAPK Fus3 (Hao *et al.*, 2008). Gβγ-bound Ste5, which is generated only when the mating pheromones are present, not only tethers the three kinases (Ste11, Ste7, and Fus3) together, but converts Fus3 to a better substrate of Ste7 (Good *et al.*, 2009, Zalatan *et al.*, 2012). In the absence of Gβγ-bound Ste5, activated Ste7 cannot phosphorylate Fus3 efficiently. Thus, there is a clear parallel between the Fus3 activation by pheromones and Hog1 activation by osmostress, namely that these kinases are converted by their respective stimulus to a better substrate of their upstream MAP2K (Fig EV5B)."

4. In Figure 5G, the negative values plotted for Hog1ΔL16 cannot be correct; the lowest possible value is zero. Something must be wrong with the calculation or normalization procedure, or perhaps with the method of background subtraction. Presumably the Hog1-P signal was undetectable, and hence it should be zero, not negative.

Response: In this experiment (now Fig 6G), both background signal (at 0M NaCl) and stimulated signal were very low (almost undetectable). Their differences from zero are insignificant statistically, and probably also biologically. However, to be consistent with other figures, we subtracted the background value from all the stimulated signals. We believe it is better this way than arbitrarily replacing negative values by zero.

5. Page 14, middle: *"... indicating that mono-phosphorylated Pbs2 could phosphorylate Hog1 (Figure 5B, lanes 3-6)."* Here it would be useful to add a direct comment that the LEVEL of Hog1-P is not equal to WT for either Pbs2 mutant (i.e., WT max = 50-60%; T518A max = 30%; S514A max = 9%).

Response: In the description of the figure (now Fig 6B), we added the following brief comment (page 14).

"However, Pbs2-WT phosphorylated Hog1 more efficiently than either S514A or T518A did, likely because Pbs2-WT could be di-phosphorylated."

Other minor points.

6. *For all of the dose-response plots throughout the paper, the zero molar NaCl data point is excluded from the graphs, despite being included in all the western blots. Why is this? I could not think of a good reason to exclude it, and I found no explanation. In some cases (e.g., Fig 3G, 3I) it partly obscures the fact that the low-osmolarity values do indeed reflect an induced response rather than a basal level.*

Response: We did not include the zero molar point in the graphs only because the lines tended to be crowded at the origin of the graph. This problem has been corrected in all the figures.

7. *Page 14, top: "Comparisons of Figure 4G with 3G, and Figure 4H with 3I, suggested that only the conditions that strongly generated phosphorylated S514 could activate Hog1 without osmotic priming." This description is a bit confusing. It would help to provide a clear and explicit description of what key feature readers should examine when making these "comparisons". Based on the parenthetical comment in the subsequent sentence, I suspect the authors wish to draw attention to the fact that S514 phosphorylation occurs only in *ste11Δ* strains and at low-to-medium osmolarity, and that these are also conditions in which the Δ L16 mutation does not compromise Hog1 phosphorylation. Is that correct? If so, perhaps state this directly.*

Response: We agree that the commented paragraph was confusing. We have change it to the following statement (page 12).

"In Fig 4, we showed that phosphorylation of Hog1 by the SLN1 branch is much more resistant to the inhibition of osmotic enhancement than that by the SHO1 branch. In Fig 5, we showed that Pbs2 is phosphorylated both at S514 and T518 by the SLN1 branch, whereas Pbs2 is phosphorylated only at T518 by the SHO1 branch. Thus, the enhancement-independent Hog1 activation by the SLN1 branch might be due to either the presence of S514-phosphorylated Pbs2 or that of di-phosphorylated Pbs2."

We hope this statement is somewhat less confusing.

8. *Page 15, line 7: "...Ste11-Q301P (which produces only mono-phosphorylated Pbs2)". I could not find where this claim was demonstrated. Perhaps this is being inferred based on the finding in Fig 4 that Ste11-WT in *ssk2/22Δ* mutants does not phosphorylate S514 of Pbs2. If so, it would be useful to clarify that this is the basis of the claim.*

Response: We changed the statement as follows (page 15).

"... Ste11-Q301P (which mimics the activation of the SHO1 branch, and likely produces only mono-phosphorylated Pbs2), "

9. The Fig 6A (now Fig 7A) legend needs some description of what is represented by each set of 3 lanes. I suspect these are independent samples done in triplicate.

Response: They are independent triplicates. We added the following statement in the main text (page 15-16).

"(Fig 7A, lanes 1-3 and 10-12, each set is an independent triplicate)"

10. Page 15, 3rd line from bottom: typo - "therefor"

Response: This typo was corrected to "therefore" (page 18)

2nd Editorial Decision

19th Dec 2019

Thank you for submitting a revised version of your manuscript. It has now been seen by one of the original referees, who finds that their main concerns have been addressed and is now in favour of publication of the manuscript. There now remain only a few editorial issues that have to be addressed before I can extend formal acceptance of the manuscript.

1. Please address the minor comment from reviewer #2.

 REFEREE REPORTS:

Referee #2:

In this revised manuscript, the authors have addressed all of my (relatively minor) prior comments in a fully satisfactory way. In addition, I have reviewed the comments from the other referee and the resultant responses and additional data provided by the authors, and I found the efforts to substantial and thorough. In my opinion the authors have fully addressed all of the issues raised. In particular, the new data (now in Figure 9) showing the temporal dynamics of Hog1 phosphorylation, in several different strains, provides several new insights into the response behaviors as well as the roles of feedback contributors. Also, the Discussion section has been enhanced significantly by the inclusion of several new topics, such as comparison to other pathways and the hypothesized evolution of Hog1 activation (as diagrammed in Fig EV5F).

Overall, I am completely satisfied by the revisions and I enthusiastically recommend publication.

I have only one small final suggestion, regarding the new results in Figure 9. In panel 9F, the feedback arrows (positive and negative) are shown as regulating the "osmosensors". It might be helpful to readers to explicitly mention in the Figure Legend that these feedback targets can include not only the traditional transmembrane osmosensors but also the "downstream osmosensor" that is the subject of this manuscript. (Indeed, the latter seems necessary to explain why the strain in Fig 9B shows both the dampening behavior and the "delayed activation" behavior, despite lacking input from transmembrane osmosensors.)

2nd Revision - authors' response

22nd Dec 2019

Point-by-point responses to the reviewers' comments

Referee #2:

In this revised manuscript, the authors have addressed all of my (relatively minor)

prior comments in a fully satisfactory way. In addition, I have reviewed the comments from the other referee and the resultant responses and additional data provided by the authors, and I found the efforts to be substantial and thorough. In my opinion the authors have fully addressed all of the issues raised. In particular, the new data (now in Figure 9) showing the temporal dynamics of Hog1 phosphorylation, in several different strains, provides several new insights into the response behaviors as well as the roles of feedback contributors. Also, the Discussion section has been enhanced significantly by the inclusion of several new topics, such as comparison to other pathways and the hypothesized evolution of Hog1 activation (as diagrammed in Fig EV5F).

Overall, I am completely satisfied by the revisions and I enthusiastically recommend publication.

Response: We thank the referee for his/her enthusiastic support for our manuscript. We have addressed the final point as described below.

Specific points.

I have only one small final suggestion, regarding the new results in Figure 9. In panel 9F, the feedback arrows (positive and negative) are shown as regulating the "osmosensors". It might be helpful to readers to explicitly mention in the Figure Legend that these feedback targets can include not only the traditional transmembrane osmosensors but also the "downstream osmosensor" that is the subject of this manuscript. (Indeed, the latter seems necessary to explain why the strain in Fig 9B shows both the dampening behavior and the "delayed activation" behavior, despite lacking input from transmembrane osmosensors.)

Response: This is indeed an important suggestion. We modified Fig 9F as suggested, and added the following statement to its legend.

"The upstream osmosensors are the traditional transmembrane osmosensors Sln1 and Sho1. The molecular identity of the downstream osmosensor, which enhances the signaling between Pbs2 and Hog1, is currently unknown."

Thank you for submitting the revised version of your manuscript. The remaining issues have now been addressed and I am now pleased to inform you that your manuscript has been accepted for publication.

YOU MUST COMPLETE ALL CELLS WITH A PINK BACKGROUND ↓
PLEASE NOTE THAT THIS CHECKLIST WILL BE PUBLISHED ALONGSIDE YOUR PAPER

Corresponding Author Name: Haruo Saito
Journal Submitted to: The EMBO Journal
Manuscript Number: EMBOJ-2019-103444